# TERMINAL VELOCITY MATCHING

**Linqi Zhou**
Luma AI

**Mathias Parger**
Luma AI

**Ayaan Haque**
Luma AI

**Jiaming Song**
Luma AI

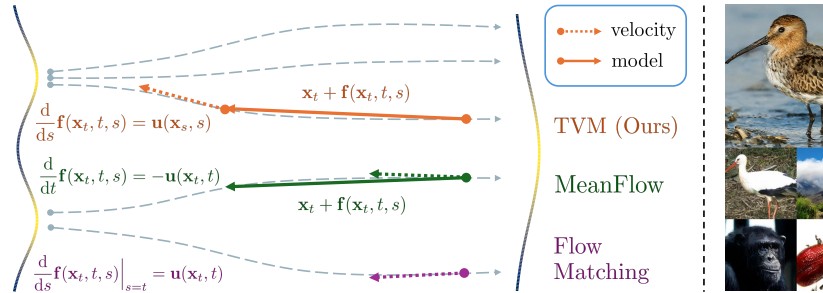

Figure 1: (*Left*) a conceptual comparison of our method to prior methods. TVM guides the one-step model via *terminal* velocity rather than *initial* velocity. (*Right*) 1-NFE samples on ImageNet at 256 and 512 resolution.

## ABSTRACT

We propose Terminal Velocity Matching (TVM), a generalization of flow matching that enables high-fidelity one- and few-step generative modeling. TVM models the transition between any two diffusion timesteps and regularizes its behavior at its terminal time rather than at the initial time. We prove that TVM provides an upper bound on the 2-Wasserstein distance between data and model distributions when the model is Lipschitz continuous. However, since Diffusion Transformers lack this property, we introduce minimal architectural changes that achieve stable, single-stage training. To make TVM efficient in practice, we develop a fused attention kernel that supports backward passes on Jacobian-Vector Products, which scale well with transformer architectures. On ImageNet-256×256, TVM achieves 3.29 FID with a single function evaluation (NFE) and 1.99 FID with 4 NFEs. It similarly achieves 4.32 1-NFE FID and 2.94 4-NFE FID on ImageNet-512×512, representing state-of-the-art performance for one/few-step models from scratch.[1]

## 1 INTRODUCTION

*Can we build generative models that simultaneously deliver high-quality samples, fast inference, and scalability to high-dimensional data, all from a single training stage?* This is the central challenge that continues to drive research in generative models. While Diffusion Models (Sohl-Dickstein et al., 2015; Ho et al., 2020; Song et al., 2020) and Flow Matching (Liu et al., 2022; Lipman et al., 2022) have become the dominant paradigms for generating images (Rombach et al., 2022; Podell et al., 2023; Esser et al., 2024) and videos (OpenAI, 2024; Wan et al., 2025), they typically require many sampling steps (*e.g.*, 50) to produce high-quality outputs. This multi-step nature makes generation computationally expensive, especially for high-dimensional data like videos.

In pursuing a single-stage training for few-step inference, recent methods have focused on directly learning integrated trajectories rather than relying on ODE solvers. Consistency-based approaches (CT (Song et al., 2023), CTM (Kim et al., 2023), sCT (Lu & Song, 2024)) and trajectory matching methods like MeanFlow (Geng et al., 2025) learn to predict or match trajectory derivatives. However, these methods lack explicit connections to distribution matching, a fundamental measure of generative model quality. While Inductive Moment Matching (IMM) (Zhou et al., 2025) addresses

---

[1]For Text-to-Image results at 10B+ scale, visit `https://lumalabs.ai/blog/engineering/tvm`

this gap by providing distribution-level guarantees through Maximum Mean Discrepancy, it requires multiple particles per training step, limiting scalability.

We propose Terminal Velocity Matching (TVM), a new framework for learning ground-truth trajectories of flow-based models in a single training stage. Instead of matching time derivatives at the initial time, TVM matches them at the *terminal* time of trajectories. This conceptually simple shift yields powerful theoretical guarantees. We prove that our training objective upper bounds the 2-Wasserstein distance between data and model distributions. Unlike IMM, our method provides distribution-level guarantees without requiring multiple particles. Our analysis also reveals a critical architectural limitation: current diffusion transformers (Peebles & Xie, 2023) lack Lipschitz continuity, which destabilizes TVM training. We address this with minimal architectural modifications, including RMSNorm-based QK-normalization and time embedding normalization.

To make TVM practical at scale, we develop an efficient Flash Attention kernel that supports backward passes on Jacobian-Vector Products (JVP), crucial for our terminal velocity computation. Our implementation achieves up to 65% speedup and significant memory reduction compared to standard PyTorch operations. We introduce a scaled parameterization where the network output naturally scales with the CFG weight $w$, allowing the model to handle varying guidance strengths more effectively. During training, we randomly sample CFG weights and directly incorporate them into our objective function with appropriate weighting ($1/w^2$) to prevent gradient explosion. This approach enables stable training across diverse guidance scales without requiring curriculum learning or specialized loss modifications, making TVM straightforward to implement and scale.

TVM achieves state-of-the-art results on ImageNet-256×256, with 3.29 FID in single-step generation (outperforming MeanFlow's (Geng et al., 2025) with 3.43 FID) and matches/exceeds diffusion baselines with just 4 function evaluation steps (*i.e.*, 1.99 FID for TVM vs. 2.27 FID for DiT). Similarly, our method surpasses diffusion baselines with 4-NFE on ImageNet-512×512 (*i.e.* 2.94 FID for TVM vs. 3.04 FID for DiT) while outperforming prior from-scratch methods such as sCT (Lu & Song, 2024) and MeanFlow on single-step generation. Our method naturally interpolates between one-step and multi-step sampling without retraining, requires no training curriculum or loss modifications, and maintains stability with simple architectures. Our construction provides new insights into building scalable one/few-step generative models with distributional guarantees, demonstrating that principled theoretical design can lead to practical improvements in both training stability and generation quality.

## 2 PRELIMINARIES: FLOW MATCHING

For a given data distribution $p_0(\mathbf{x}_0)$ and prior distribution $p_1(\mathbf{x}_1)$, Flow Matching (FM) (Lipman et al., 2022; Liu et al., 2022) constructs a time-augmented linear interpolation $\mathbf{x}_t$ between data $\mathbf{x}_0 \in \mathbb{R}^D$ and prior $\mathbf{x}_1 \in \mathbb{R}^D$ such that $\mathbf{x}_t = (1-t)\mathbf{x}_0 + t\mathbf{x}_1$[2]. For each path $\mathbf{x}_t$ conditioned on a $(\mathbf{x}_0, \mathbf{x}_1)$ pair, there exists a conditional velocity $\mathbf{v}_t = \mathbf{x}_1 - \mathbf{x}_0$ for each $\mathbf{x}_t$. Under this definition, a ground-truth velocity field $\mathbf{u} : \mathbb{R}^D \times [0,1] \to \mathbb{R}^D$ marginal over all data and prior exists but is not known in analytical form. Therefore, a neural network $\mathbf{u}_\theta(\mathbf{x}_t, t)$ is used as approximation via loss

$$\mathcal{L}_{\text{FM}}(\theta) = \mathbb{E}_{\mathbf{x}_t, \mathbf{v}_t, t}[\|\mathbf{u}_\theta(\mathbf{x}_t, t) - \mathbf{v}_t\|_2^2] \tag{1}$$

for all $t \in [0,1]$ and $\mathbf{x}_t \sim p_t(\mathbf{x}_t)$ where $p_t(\mathbf{x}_t)$ denotes the marginal distribution over all data and prior. It can be shown that the minimizer $\theta_{\min}$ implies $\mathbf{u}_{\theta_{\min}}(\mathbf{x}_t, t) = \mathbf{u}(\mathbf{x}_t, t)$ which can be used during inference to transport prior to data distribution by solving an ODE $\frac{\mathrm{d}}{\mathrm{d}t}\mathbf{x}_t = \mathbf{u}(\mathbf{x}_t, t)$.

For each ground-truth $\mathbf{u}(\mathbf{x}_t, t)$, there exists a corresponding displacement map $\psi : \mathbb{R}^D \times [0,1] \times [0,1] \to \mathbb{R}^D$ (*i.e.* flow map (Boffi et al., 2024)) from any start time $t \in [0,1]$ to an end time $s \in [0,1]$. It is defined as the ODE integral following $\mathbf{u}(\mathbf{x}_r, r)$ for all $r \in [s, t]$, *i.e.*

$$\psi(\mathbf{x}_t, t, s) = \mathbf{x}_t + \int_t^s \mathbf{u}(\mathbf{x}_r, r)\mathrm{d}r. \tag{2}$$

Empirically, $\mathbf{u}_\theta(\mathbf{x}_t, t)$ is used with classical ODE integration techniques such as the Euler method to produce samples.

---

[2]See Lipman et al. (2022); Albergo et al. (2023) for general path constructions.

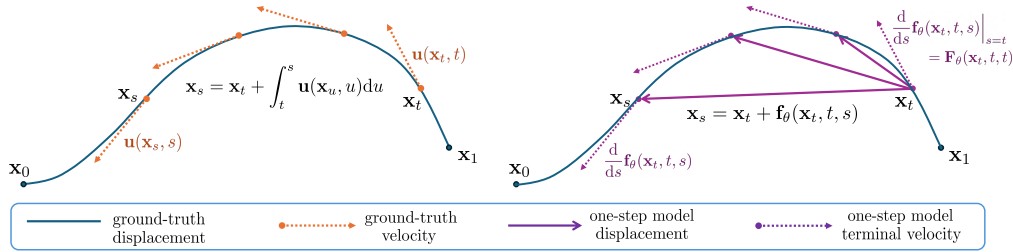

Figure 2: An illustration of Terminal Velocity Matching. Left shows the ground-truth displacement map by integrating the true velocity. Right shows our model path directly jumping between points on the ground-truth path in one step. In our method, the one-step generation $\mathbf{x}_0$ from $\mathbf{x}_t$ coincides with ground-truth $\mathbf{x}_0$ if the terminal velocity of model $\frac{\mathrm{d}}{\mathrm{d}s}\mathbf{f}(\mathbf{x}_t, t, s)$ coincides with ground-truth velocity $\mathbf{u}(\mathbf{x}_s, s)$ for all $s \in [0, t]$ along the true flow path (see Eq. (7)). The terminal velocity condition is jointly satisfied with the boundary case when model displacement is 0, where matching $\frac{\mathrm{d}}{\mathrm{d}s}\mathbf{f}(\mathbf{x}_t, t, s)|_{s=t}$ with $\mathbf{u}(\mathbf{x}_t, t)$ reduces to Flow Matching.

## 3 TERMINAL VELOCITY MATCHING

We propose Terminal Velocity Matching (TVM), a single-stage objective that directly learns the ODE integral in Eq. 2. By learning the transition between any two timesteps, TVM can generate high quality solutions in one step or few steps, while enjoying inference-time scaling.

Let $\mathbf{f}(\mathbf{x}_t, t, s) := \psi(\mathbf{x}_t, t, s) - \mathbf{x}_t$ denote the *net* displacement of the velocity field. We observe that it must satisfy the following two conditions:

$$\textcircled{1} \quad \mathbf{f}(\mathbf{x}_t, t, s) = \int_t^s \mathbf{u}(\mathbf{x}_r, r)\mathrm{d}r \,, \qquad \textcircled{2} \quad \frac{\mathrm{d}}{\mathrm{d}s}\mathbf{f}(\mathbf{x}_t, t, s)\Big|_{s=t} = \mathbf{u}(\mathbf{x}_t, t). \qquad (3)$$

The first condition is the definition of net displacement and the second condition is true by differentiating both sides of the first condition w.r.t. $s$ evaluated at $s = t$. It explicitly relates the displacement map (with large time jump) to the marginal velocity field (with infinitesimal time jump), allowing us to interpolate between one-step sampling and ODE-like infinite-step sampling.

One of our key insights is that we can use a single two-time conditioned neural network $\mathbf{F}_\theta(\mathbf{x}_t, t, s)$ to learn both the one-step displacement sampler from $t$ to $s$ and the instantaneous velocity field. For simplicity, we let our model with learnable parameters $\theta$ be

$$\mathbf{f}_\theta(\mathbf{x}_t, t, s) = (s - t)\mathbf{F}_\theta(\mathbf{x}_t, t, s), \qquad \mathbf{u}_\theta(\mathbf{x}_t, t) := \frac{\mathrm{d}}{\mathrm{d}s}\mathbf{f}_\theta(\mathbf{x}_t, t, s)\Big|_{s=t} = \mathbf{F}_\theta(\mathbf{x}_t, t, t) \qquad (4)$$

where the scaling $(s - t)$ is chosen to satisfy integral boundary condition when $t = s$[3]. Condition $\textcircled{2}$ can be easily enforced by FM loss (in Eq. (1)) and condition $\textcircled{1}$ can be naïvely enforced via the *displacement error*

$$\mathcal{L}_{\mathrm{displ}}^t(\theta) := \mathbb{E}_{\mathbf{x}_t}\left[\left\|\mathbf{f}_\theta(\mathbf{x}_t, t, 0) - \int_t^0 \mathbf{u}(\mathbf{x}_r, r)\mathrm{d}r\right\|_2^2\right]. \qquad (5)$$

Once the above error is minimized to zero, one can obtain one-step samples by calling $\mathbf{x}_t + \mathbf{f}_\theta(\mathbf{x}_t, t, 0)$ for any $\mathbf{x}_t \sim p_t(\mathbf{x}_t)$ at $t \in [0, 1]$. However, this objective is infeasible because it requires ODE integration for each starting point $\mathbf{x}_t$. We address this challenge by proposing a simple sufficient condition to the network that bypasses explicit training-time ODE simulation.

**Terminal Velocity Condition.** Explicit integration can be bypassed via differentiating w.r.t. integral boundaries. For the ground-truth net displacement $\mathbf{f}(\mathbf{x}_t, t, s)$ in condition $\textcircled{1}$, differentiating w.r.t. $s$ gives rise to the following condition on terminal velocity, *i.e.*

$$\frac{\mathrm{d}}{\mathrm{d}s}\mathbf{f}(\mathbf{x}_t, t, s) = \mathbf{u}(\psi(\mathbf{x}_t, t, s), s). \qquad (6)$$

This condition is true for any ground-truth net displacement $\mathbf{f}$, and we show in Appendix A.2 that given $t \in [0, 1]$ and our parameterized map $\mathbf{f}_\theta(\mathbf{x}_t, t, s)$,

$$\mathcal{L}_{\mathrm{displ}}^t(\theta) \leq \int_0^t \mathbb{E}_{\mathbf{x}_t}\left[\left\|\frac{\mathrm{d}}{\mathrm{d}s}\mathbf{f}_\theta(\mathbf{x}_t, t, s) - \mathbf{u}(\psi(\mathbf{x}_t, t, s), s)\right\|_2^2\right]\mathrm{d}s. \qquad (7)$$

---

[3]This is similar to CTM (Kim et al., 2023). See Appendix A.1 for conditions on general scaling factors.

This result shows that the *terminal velocity error* on the right hand side upper bounds the displacement error, and so zero terminal velocity error implies that displacement from $t$ to $0$ matches exactly. Moreover, it is easy to see that the terminal velocity error reduces to the marginal FM loss as $t \to s$ (see Appendix A.3). FM can thus be understood as matching a trajectory's terminal velocity when the net displacement is $0$. An illustration of our framework is shown in Figure 2. Despite the simplicity and generality, in practice, fulfilling this condition is still difficult due to the requirement of $\psi$ and $\mathbf{u}$. Fortunately, this issue can be effectively addressed using learned network as *proxies*.

**Learned networks as proxies.** Specifically, we propose the following approximation

$$\mathbf{u}(\psi(\mathbf{x}_t, t, s), s) \approx \mathbf{u}_\theta(\mathbf{x}_t + \mathbf{f}_\theta(\mathbf{x}_t, t, s), s) \tag{8}$$

as proxies for the ground-truths. To properly guide the terminal velocity, $\mathbf{u}_\theta(\mathbf{x}_s, s)$ needs to first approximate the ground-truth $\mathbf{u}(\mathbf{x}_s, s)$ for any $\mathbf{x}_s$ and $s$. Therefore, the *proxy terminal velocity error* can be jointly optimized with Flow Matching, which, as noted above, is a special boundary case of the terminal velocity error when displacement is $0$. We use the term "Terminal Velocity Matching" for this joint minimization of general and boundary-case velocity error, where the objective is

$$\mathcal{L}_{\text{TVM}}^{t,s}(\theta) = \mathbb{E}_{\mathbf{x}_t, \mathbf{x}_s, \mathbf{v}_s} \left[ \underbrace{\left\| \frac{\mathrm{d}}{\mathrm{d}s} \mathbf{f}_\theta(\mathbf{x}_t, t, s) - \mathbf{u}_\theta(\mathbf{x}_t + \mathbf{f}_\theta(\mathbf{x}_t, t, s), s) \right\|_2^2}_{\text{satisfies } \textcircled{1}} + \underbrace{\left\| \mathbf{u}_\theta(\mathbf{x}_s, s) - \mathbf{v}_s \right\|_2^2}_{\text{satisfies } \textcircled{2}} \right] \tag{9}$$

for each time $t \in [0, 1]$ and $s \in [0, t]$. Intuitively, this objective leverages a single network to parameterize both the instantaneous velocity field and the displacement map, the former of which is learned from data to guide the learning of the latter. To provide further theoretical justification, in the following theorem, we formally establish a weighted integral of our objective as a proper upper bound on the 2-Wasserstein distance between the data distribution $p_0(\mathbf{x}_0)$ and our model distribution $\mathbf{f}_{t\to0}^\theta \# p_t(\mathbf{x}_t)$ pushforward from $p_t(\mathbf{x}_t)$ via our parameterized flow map.

**Theorem 1** (Connection to the 2-Wasserstein distance). *Given $t \in [0, 1]$, let $\mathbf{f}_{t\to0}^\theta \# p_t(\mathbf{x}_t)$ be the distribution pushforward from $p_t(\mathbf{x}_t)$ via $\mathbf{f}_\theta(\mathbf{x}_t, t, 0)$, and assume $\mathbf{u}_\theta(\cdot, s)$ is Lipschitz-continuous for all $s \in [0, t]$ with Lipschitz constants $L(s)$, with additional mild regularity conditions,*

$$W_2^2(\mathbf{f}_{t\to0}^\theta \# p_t, p_0) \leq \int_0^t \lambda[L](s) \mathcal{L}_{\text{TVM}}^{t,s}(\theta) \mathrm{d}s + C, \tag{10}$$

*where $W_2(\cdot, \cdot)$ is 2-Wasserstein distance, $\lambda[\cdot]$ is a functional of $L(\cdot)$, and $C$ is a non-optimizable constant.*

**Training objective.** The theorem relates our per-time objective to distribution divergence. However, for practicality, we avoid computation of the above weighting function and instead choose to randomly sample both $t$ and $s$ via distribution $p(s, t)$ such that

$$\mathcal{L}_{\text{TVM}}(\theta) = \mathbb{E}_{t,s} \left[ \mathcal{L}_{\text{TVM}}^{t,s}(\theta) \right] \tag{11}$$

where notably $\mathcal{L}_{\text{TVM}}(\theta)$ reduces to Flow Matching objective when $t = s$ (see Appendix A.5). In practice, we employ a biased estimate of the above objective by using exponentially averaged (EMA) weights and stop-gradient for our proxy networks (Li et al., 2023). The biased per-time objective $\hat{\mathcal{L}}_{\text{TVM}}^{t,s}(\theta)$ is

$$\mathbb{E}_{\mathbf{x}_t, \mathbf{x}_s, \mathbf{v}_s} \left[ \left\| \frac{\mathrm{d}}{\mathrm{d}s} \mathbf{f}_\theta(\mathbf{x}_t, t, s) - \mathbf{u}_{\theta_{\text{sg}}^*}(\mathbf{x}_t + \mathbf{f}_{\theta_{\text{sg}}}(\mathbf{x}_t, t, s), s) \right\|_2^2 \mathbb{1}_{t \neq s} + \left\| \mathbf{u}_\theta(\mathbf{x}_s, s) - \mathbf{v}_s \right\|_2^2 \right] \tag{12}$$

where $\theta_{\text{sg}}$ and $\theta_{\text{sg}}^*$ are the stop-grad weight and stop-grad EMA weight of $\theta$, and $\mathbb{1}_{t \neq s}$ is $0$ when $t = s$ and $1$ otherwise to ensure the constraint to reduce to FM loss when $t = s$.

**Classifier-free guidance (CFG).** In the case of class-conditional generation. The ground-truth velocity field is replaced by a linear combination of class-conditional velocity $\mathbf{u}(\mathbf{x}_r, r, c)$ and unconditional velocity $\mathbf{u}(\mathbf{x}_r, r)$ (Ho & Salimans, 2022), such that the new displacement map is

$$\psi_w(\mathbf{x}_t, t, s, c) = \mathbf{x}_t + \int_t^s [w\mathbf{u}(\mathbf{x}_r, r, c) + (1 - w)\mathbf{u}(\mathbf{x}_r, r)] \,\mathrm{d}r, \tag{13}$$

where $w$ is the CFG weight, $c$ is class and $\varnothing$ denotes empty label. To train with CFG, we additionally condition the network on $w$ and $c$, and our class-conditional map is $\mathbf{f}_\theta(\mathbf{x}_t, t, s, c, w) = (s-t)\mathbf{F}_\theta(\mathbf{x}_t, t, s, c, w)$ where the additional $w$ scale is chosen due to linear scaling in magnitude for marginal velocity w.r.t. $w$. The instantaneous velocity $\mathbf{u}_\theta(\mathbf{x}_s, s, c, w)$ is regressed against conditional velocity $w\mathbf{v}_t + (1-w)\mathbf{u}(\mathbf{x}_r, r)$ where we can approximate $\mathbf{u}(\mathbf{x}_r, r)$ with our own network (Chen et al., 2025). The per-time and per-class Flow Matching term can be modified as

$$\hat{\mathcal{L}}_{\mathrm{FM}}^{s,c,w}(\theta) = \mathbb{E}_{\mathbf{x}_s, \mathbf{v}_s} \left[ \left\| \mathbf{u}_\theta(\mathbf{x}_s, s, c, w) - \left[ w\mathbf{v}_s + (1-w)\mathbf{u}_{\theta_{\mathrm{sg}}^*}(\mathbf{x}_s, s, \varnothing, 1) \right] \right\|_2^2 \right], \qquad (14)$$

where $\theta_{\mathrm{sg}}^*$ denotes EMA weights. We show in Appendix A.6 that the minimizer of this objective coincides with the ground-truth CFG velocity in Eq. (13). Our class-conditional objective $\hat{\mathcal{L}}_{\mathrm{TVM}}^{t,s,w}(\theta)$ can be modified as

$$\frac{1}{w^2}\mathbb{E}_{\mathbf{x}_t, c}\left[ \left\| \frac{\mathrm{d}}{\mathrm{d}s}\mathbf{f}_\theta(\mathbf{x}_t, t, s, c, w) - \mathbf{u}_{\theta_{\mathrm{sg}}^*}(\mathbf{x}_t + \mathbf{f}_{\theta_{\mathrm{sg}}}(\mathbf{x}_t, t, s, c, w), s, c, w) \right\|_2^2 \mathbb{1}_{t\neq s} + \hat{\mathcal{L}}_{\mathrm{FM}}^{s,c,w}(\theta) \right]. \quad (15)$$

The weighting $1/w^2$ is to prevent exploding gradients because the magnitude of ground-truth velocity scales linearly with $w$. Final objective simply samples each of $t, s, w$ under some distribution $p(t, s)p(w)$ and computes the above loss in expectation. We randomly set $c = \varnothing$ with some probability (e.g. 10%) and for each $c = \varnothing$ we set $w = 1$. Our training algorithm is shown in Algorithm 1.

**Sampling.** Our construction can naturally interpolate between one-step and $n$-step sampling. See Figure 3 for PyTorch-style sampling code.

```python
def sampling(net, x, n, c, w):
    ts = torch.linspace(1, 0, n+1)
    for t,s in zip(ts[:1],ts[1:]):
        x = x + (s-t) * net(x, t, s, c, w)
    return x
```

Figure 3: PyTorch-style sampling code.

## 4 PRACTICAL CHALLENGES

We note and address several challenges to practically implement our objective.

**Semi-Lipschitz control.** Theorem 1 makes the crucial assumption that $\mathbf{u}_\theta(\mathbf{x}_s, s)$ is Lipschitz continuous. However, modern transformers with scaled dot-product attention (SDPA) and LayerNorm (LN, Ba et al. (2016)) are *not* Lipschitz continuous (Kim et al., 2021; Qi et al., 2023; Castin et al., 2023). This issue similarly applies to diffusion transformers (DiT) (Peebles & Xie, 2023). Our insight is to make minimal and non-restrictive changes to the architecture for Lipschitz control.

As shown in Figure 4, the original DiT experiences training instability leading to steep jump in network activations. As a solution, we adopt RMSNorm as QK-Norm, which coincides with the proposed $\mathcal{L}_2$ QK-Norm (Qi et al., 2023) with learnable scaling and is provably Lipschitz continuous. We also substitute all LN with RMSNorm (without learnable parameters, denoted as RMSNorm$^-(\cdot)$), whose Lipschitzness we show in Appendix B.1. In addition, DiT introduces Adaptive LayerNorm (AdaLN) where the output of RMSNorm is modulated by MLP outputs of time embeddings denoted as RMSNorm$^-(x) \odot a(t) + b(t)$ where $x$ is the input feature and $a(t), b(t)$ are scale and shift respectively. However, the Lipschitz constant of this layer depends on the magnitude of $a(t)$

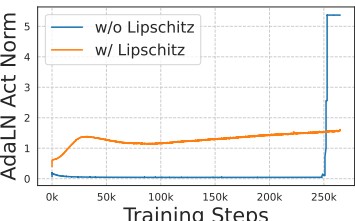

Figure 4: Activation norm of last time embedding layer. Same trends follow for all other layers.

which can grow unbounded and is subject to instability. We therefore employ RMSNorm$^-(\cdot)$ again on all modulation parameters for

$$\mathrm{AdaLN}(x, t) = \mathrm{RMSNorm}^-(x) \odot \mathrm{RMSNorm}^-(a(t)) + \mathrm{RMSNorm}^-(b(t)). \qquad (16)$$

Figure 4 also shows the activation with our proposed changes. Activations stay smooth after our fixes. Finally, we follow Qi et al. (2023) and use Lipschitz initialization for all linear layers except for time embedding layers. Note that these modifications do not explicitly constrain the Lipschitz constants of all but the key layers where instability can arise. We find such partial control of the Lipschitzness is sufficient for empirical success.

**Flash Attention JVP with backward pass.** The training objective involves the time derivative of our map $\mathbf{f}_\theta(\mathbf{x}_t, t, s)$, which can be derived as

$$\frac{\mathrm{d}}{\mathrm{d}s}\mathbf{f}_\theta(\mathbf{x}_t, t, s) = \mathbf{F}_\theta(\mathbf{x}_t, t, s) + (s-t)\partial_s\mathbf{F}_\theta(\mathbf{x}_t, t, s) \tag{17}$$

where the last term involves differentiating through the network with Jacobian-Vector Product (JVP). This poses significant challenge for transformers because automatic differentiation packages, *e.g.* PyTorch, often do not efficiently handle JVP of SDPA. Open-source Flash Attention (Dao et al., 2022) also has limited support for JVP. Crucially, different from prior works (Lu & Song, 2024; Geng et al., 2025; Sabour et al., 2025), gradient is also propagated through the JVP term $\partial_s\mathbf{F}_\theta(\mathbf{x}_t, t, s)$. To tackle these challenges, we propose an efficient Flash Attention kernel that (i) fuses JVP with forward pass, (ii) uses significantly less memory than naïve PyTorch attention, and (iii) supports backward pass on JVP results. We detail the implementation in Appendix C.

**Optimizer parameter change.** Due to higher-order gradient through JVP, our loss can be subject to fluctuation with the default AdamW $\beta_2 = 0.999$. We take inspiration from language models (Touvron et al., 2023) for mitigation and use $\beta_2 = 0.95$ to speed up update of the gradient second moment. As show in Figure 5, the terminal velocity error fluctuates significantly less after $\beta_2$ change.

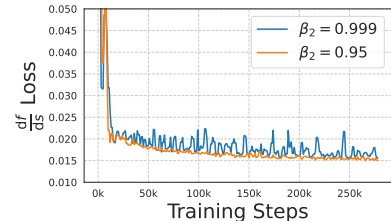

Figure 5: Smoother terminal velocity error with $\beta_2 = 0.95$.

**Scaled parameterization.** The ground-truth CFG velocity scales linearly in magnitude with $w$, so using neural networks to directly predict the velocity may be suboptimal. We therefore additionally investigate a simple scaled alternative as $\mathbf{f}_\theta(\mathbf{x}_t, t, s, c, w) = (s-t)w\mathbf{F}_\theta(\mathbf{x}_t, t, s, c, w)$ so that $\mathbf{u}_\theta(\mathbf{x}_s, s, c, w) = w\mathbf{F}_\theta(\mathbf{x}_s, s, s, c, w)$ which scales with $w$ by design. We study the effect of this parameterization in experiments.

**Different time distribution for FM loss.** We find it empirically helpful to use a separate distribution to sample different $s$ specifically for the FM loss (see Eq. (15))This is because we can directly transfer the proven successful time distribution from FM training for TVM. How this is used can be found in Algorithm 1 and we ablate this decision in Section 7.3.

## 5 CONNECTION TO PRIOR WORKS

**MeanFlow.** MeanFlow (Geng et al., 2025) minimizes loss $E_{\mathbf{x}_t, t, s}\left[\|\mathbf{F}_\theta(\mathbf{x}_t, t, s) - F_{\mathrm{tgt}}\|_2^2\right]$ where

$$F_{\mathrm{tgt}} = \mathbf{u}(\mathbf{x}_t, t) + (s-t)\left[\mathbf{u}(\mathbf{x}_t, t) \cdot \nabla_{\mathbf{x}_t}\mathbf{F}_{\theta_{\mathrm{sg}}}(\mathbf{x}_t, t, s) + \partial_t\mathbf{F}_{\theta_{\mathrm{sg}}}(\mathbf{x}_t, t, s)\right] \tag{18}$$

This loss can be equivalently rewritten as $E_{\mathbf{x}_t, t, s}\left[\left\|\frac{\mathrm{d}}{\mathrm{d}t}\mathbf{f}_\theta(\mathbf{x}_t, t, s) + \mathbf{u}(\mathbf{x}_t, t)\right\|_2^2\right]$ where $\mathbf{f}_\theta(\mathbf{x}_t, t, s) = (s-t)\mathbf{F}_\theta(\mathbf{x}_t, t, s)$ and loss is minimized if and only if $\frac{\mathrm{d}}{\mathrm{d}t}\mathbf{f}_\theta(\mathbf{x}_t, t, s) = -\mathbf{u}(\mathbf{x}_t, t)$ (see Appendix E.1). This exhibits duality with our proposed method in that we enforce a differential condition w.r.t. $s$ while MeanFlow differentiates w.r.t. $t$ which requires $\mathbf{u}(\mathbf{x}_t, t)$ to be propagated through JVP. In practice, $\mathbf{u}(\mathbf{x}_t, t)$ is replaced with $\mathbf{v}_t$, which introduces additional variance during training and can cause fluctuation in gradient, especially under random CFG during training (see Section 7.2). Additionally, the relationship between the loss and distribution divergence remains elusive with the introduction of $\mathbf{v}_t$. In contrast, we show our loss upper bounds 2-Wasserstein distance, which provides unique insights on smoothness properties of our network.

**Physics Informed Distillation (PID).** PID (Tee et al., 2024) as inspired by Physics Informed Neural Networks (Raissi et al., 2019; Cuomo et al., 2022) distills pretrained diffusion models $\mathbf{u}_\phi(\mathbf{x}_t, t)$ into one-step samplers. It parameterizes the one-step net displacement as $\mathbf{f}_\theta(\mathbf{x}_1, s) = (s-1)\mathbf{u}_\theta(\mathbf{x}_1, s)$ where $\mathbf{x}_1 \sim p_1(\mathbf{x}_1)$ and trains via distillation loss

$$\mathbb{E}_{\mathbf{x}_1, s}\left[\left\|\frac{\mathrm{d}}{\mathrm{d}s}\mathbf{f}_\theta(\mathbf{x}_1, s) - \mathbf{u}_\phi(\mathbf{x}_1 + \mathbf{f}_{\theta_{\mathrm{sg}}}(\mathbf{x}_1, s), s)\right\|_2^2\right] \tag{19}$$

Our method generalizes the setting by introducing the starting time $t$ in addition to the terminal time $s$. Under this view, PID sets $t = 1$ and can only generate one-step samples. We additionally show

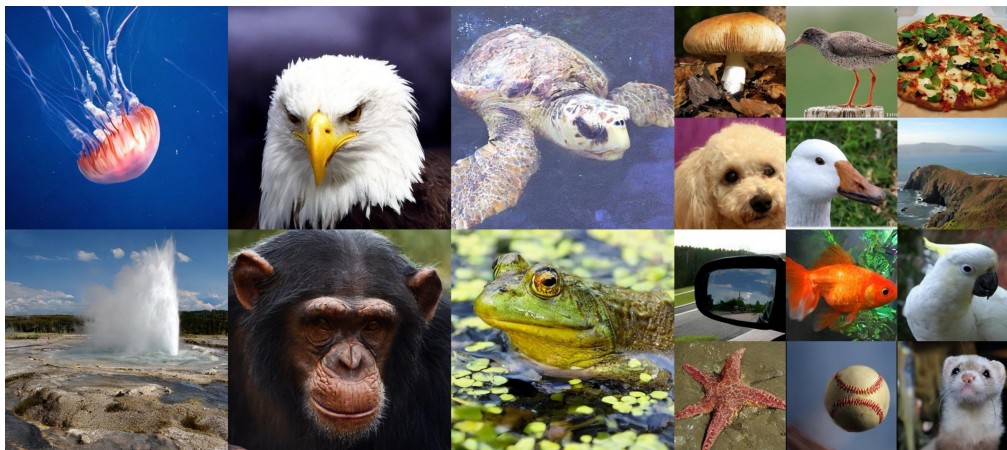

Figure 6: One-step samples from TVM on both ImageNet-512×512 and ImageNet-256×256.

in Section 7.3 that naïve combination of PID and FM loss suffers from optimization instability and a continuous distribution on $t$ is necessary for empirical success.

## 6 RELATED WORKS

**Diffusion and Flow Matching.** Diffusion models (Sohl-Dickstein et al., 2015; Ho et al., 2020; Song et al., 2020) learn generative models by reversing stochastic processes, while Flow Matching (Liu et al., 2022; Lipman et al., 2022) generalizes this to arbitrary priors with simplified training. Both approaches ultimately solve ODEs with neural networks during sampling.

**One-Step and Few-Step Models from Scratch.** To address slow inference from ODE simulation, recent methods aim for few-step generation in a single training stage. Consistency models (Song et al., 2023; Lu & Song, 2024) parameterize networks to represent ODE integrals but cannot jump between arbitrary timesteps without injecting additional noise, which can limit multi-step performance. Two-time conditioned approaches enable arbitrary timestep transitions: IMM (Zhou et al., 2025) provides distribution consistency via Maximum Mean Discrepancy but requires multiple particles; MeanFlow (Geng et al., 2025) and Flow Map Matching (Boffi et al., 2024) match trajectory derivatives but lack distributional guarantees. Other variants bypass differentiation via Monte Carlo (Liu & Yue, 2025) or combine distillation with FM (Frans et al., 2024).

Unlike these methods, TVM regularizes path behavior at the *terminal time* rather than initial time and provides explicit 2-Wasserstein bounds. While sCT and MeanFlow only compute forward JVP, TVM uniquely supports backward passes through the JVP computation, These innovations drive both our theoretical insights and architectural improvements. Another recent work (Boffi et al., 2025) has proposed a unifying perspective on training flow maps from scratch; it encompasses many approaches such as Shortcut (Frans et al., 2024), CM (Song et al., 2023; Lu & Song, 2024), MeanFlow (Geng et al., 2025) and shares similar theoretical insights as TVM.

## 7 EXPERIMENTS

We investigate how well TVM can generate natural images (Section 7.1), discuss its advantages compared to previous methods (Section 7.2), ablate various practical choices (Section 7.3) and discuss its computation cost (Section 7.4).

### 7.1 IMAGE GENERATION

**ImageNet-256×256.** We present quantitative results in Table 1 under FID (Heusel et al., 2017). We adopt the default DiT-XL/2 architecture (Peebles & Xie, 2023) and inject $t - s$ as the second timestep, following IMM (Zhou et al., 2025) and MeanFlow (Geng et al., 2025). We additionally employ our semi-Lipschitz control techniques for training stability or we notice activation explosion as described in Figure 4, and we train with constantly sampled CFG, *i.e.* models with $w = 2$

| Diffusion/Flow | NFE (↓) | FID (↓) | # Params. |
|---|---|---|---|
| ADM (Dhariwal & Nichol, 2021) | 250×2 | 10.96 | 554M |
| LDM-4-G (Rombach et al., 2022) | 250×2 | 3.60 | 400M |
| DiT-XL/2 (Peebles & Xie, 2023) ($w = 1.25$) | 250×2 | 3.22 | 675M |
| DiT-XL/2 (Peebles & Xie, 2023) ($w = 1.5$) | 250×2 | 2.27 | 675M |
| SiT-XL/2 (Ma et al., 2024) ($w = 1.5$) | 250×2 | 2.15 | 675M |
| One/Few-Step from Scratch | | | |
| iCT-XL/2 (Song & Dhariwal, 2023) | 1 | 34.24 | 675M |
| | 2 | 20.3 | 675M |
| Shortcut-XL/2 (Frans et al., 2024) | 1 | 10.60 | 675M |
| IMM-XL/2 (Zhou et al., 2025) | 1 × 2 | 8.05 | 675M |
| | 2 × 2 | 3.99 | 675M |
| | 2 × 4 | 2.51 | 675M |
| MeanFlow-XL/2 (Geng et al., 2025) | 1 | 3.43 | 676M |
| | 2 | 2.93 | 676M |
| **TVM-XL/2 (Ours)** ($w = 2$) | 1 | **3.29** | 678M |
| | 2 | 2.80 | 678M |
| **TVM-XL/2 (Ours)** ($w = 1.75$) | 1 | 4.58 | 678M |
| | 2 | **2.61** | 678M |
| | 4 | **1.99** | 678M |

Table 1: FID results on ImageNet-256×256.

| Diffusion/Flow | NFE (↓) | FID (↓) | # Params. |
|---|---|---|---|
| ADM-G (Dhariwal & Nichol, 2021) | 250×2 | 7.72 | 559M |
| SimDiff (Hoogeboom et al., 2023) | 512×2 | 3.02 | 2B |
| VDM++ (Kingma & Gao, 2024) | 512×2 | 2.65 | 2B |
| U-ViT-H/4 (Bao et al., 2023) | 250×2 | 4.05 | 501M |
| EDM2-L (Karras et al., 2024) | 63×2 | 1.88 | 778M |
| EDM2-XL (Karras et al., 2024) | 63×2 | 1.85 | 1.1B |
| DiT-XL/2 (Peebles & Xie, 2023) ($w = 1.25$) | 250×2 | 4.64 | 675M |
| DiT-XL/2 (Peebles & Xie, 2023) ($w = 1.5$) | 250×2 | 3.04 | 675M |
| SiT-XL/2 (Ma et al., 2024) ($w = 1.5$) | 250×2 | 2.62 | 675M |
| One/Few-Step from Scratch | | | |
| sCT-L (Lu & Song, 2024) | 1 | 5.15 | 778M |
| | 2 | 4.65 | 778M |
| sCT-XL (Lu & Song, 2024) | 1 | 4.33 | 1.1B |
| | 2 | 3.73 | 1.1B |
| MeanFlow-XL/2 (Geng et al., 2025) | 1 | 5.24 | 676M |
| | 2 | 3.17 | 676M |
| **TVM-XL/2 (Ours)** ($w = 2.50$) | 1 | **4.32** | 678M |
| | 2 | **3.50** | 678M |
| **TVM-XL/2 (Ours)** ($w = 2.25$) | 1 | 5.37 | 678M |
| | 2 | 3.89 | 678M |
| | 4 | **2.94** | 678M |

Table 2: FID results on ImageNet-512×512.

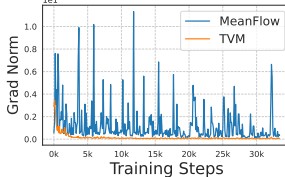 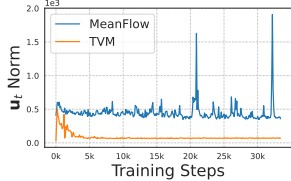 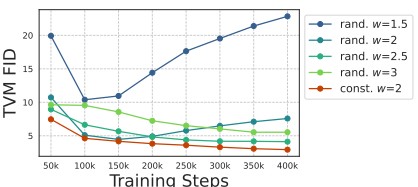

Figure 7: *(Left)* MeanFlow is subject to wide variation in gradient norm if CFG scales (i.e., $\kappa$ and $\omega$) are randomly sampled under naïve settings (see Appendix F.2 for details). TVM shows much smoother gradient norm. *(Middle)* MeanFlow's gradient norm is strongly correlated with the fluctuation of $\|\mathbf{u}(\mathbf{x}_t, t)\|$. TVM's $\|\mathbf{u}(\mathbf{x}_t, t)\|$ is much more stable under the same CFG setting. *(Right)* Our method converges with random CFG at training time, although tradeoff exists between different CFG in FID. Constantly sampled CFG works best.

and $w = 1.75$ are two different models trained from scratch. We describe additional training details in Appendix F. Our method achieves state-of-the-art 1-NFE FID among methods trained from scratch, outperforming MeanFlow and IMM. With CFG $w = 2$, TVM can achieve noticeable improvements over MeanFlow, e,g, 3.29 FID vs. 3.43 FID for 1-NFE and similarly for 2-NFE. With 4 NFEs, $w = 1.75$ also exceeds 500-NFE diffusion baselines. We additionally show qualitative 1-NFE samples on the right of Figure 6.

**ImageNet-512×512.** We train with the same settings as in 256×256-resolution and we show the FID scores in Table 2. We rerun MeanFlow using the same settings as in ImageNet-256×256 as our baseline in addition to sCT (Lu & Song, 2024) under similar model sizes. TVM again outperforms sCT and MeanFlow in 1-NFE and 2-NFE regime. Notably, TVM-XL/2 outperforms sCT-XL with 1.1B parameters, highlighting TVM's more optimal use of model capacity in fitting the image distribution. Moreover, with $w = 2.25$, TVM with 4-NFE can match 500-NFE DiT-XL/2 baseline in performance, further demonstrating the scalability of our algorithm to higher resolution.

Intriguingly, for both datasets, TVM trained with higher CFG performs better on 1 NFE while worse on 2 NFEs. We believe this implies a fundamental trade-off between different NFE quality and that the network is limited in capacity in fitting all NFEs well. We leave more detailed studies on this trade-off and any design improvements to future work.

## 7.2 DISCUSSION ON TRAINING ADVANTAGES

**Single sample objective.** Unlike IMM (Zhou et al., 2025) which uses more than 4 samples to calculate its loss, we use a single sample to for loss calculation without losing a distribution-matching interpretation. This also allows the objective to be scaled to large models and high-dimensional datasets where batch size on each GPU is constrained to be 1.

**Training with random CFG.** Our construction allows us to randomly sample CFG scale during training without collapse. We attribute this stability to our JVP being only calculated w.r.t. $s$ which is invariant to starting position $\mathbf{x}_t$ and time $t$. In contrast, CT (Song et al., 2023; Lu & Song, 2024) and MeanFlow (Geng et al., 2025) require velocity $\mathbf{u}(\mathbf{x}_t, t)$ to be used in the JVP calculation. In

| trunc $(\mu_t,\sigma_t),(\mu_s,\sigma_s)$ | FID | clamp $(\mu_t,\sigma_t),(\mu_s,\sigma_s)$ | FID | gap $(\mu_g,\sigma_g),(\mu_s,\sigma_s)$ | FID |
|---|---|---|---|---|---|
| $(-0.4,1.0),(-0.4,1.0)$ | 4.59 | $(2.0,2.0),(-0.4,1.0)$ | 3.88 | $(-0.4,1.0),(-0.4,1.0)$ | 5.12 |
| $(2.0,1.0),(-0.4,1.0)$ | 4.00 | $(2.0,1.0),(-0.4,1.0)$ | 4.11 | $(-0.8,1.0),(-0.4,1.0)$ | **3.72** |
| $(2.0,2.0),(-0.4,1.0)$ | 4.01 | $(2.0,1.0),(-0.6,1.0)$ | 4.00 | $(-0.8,1.4),(-0.4,1.0)$ | 3.95 |
| $(2.0,2.0),(-0.6,1.0)$ | 7.88 | $(1.0,1.0),(-0.4,1.0)$ | **3.66** | $(-1.0,1.2),(-0.4,1.0)$ | 3.82 |
| $(1.0,1.0),(-0.4,1.0)$ | **3.70** | $(1.0,2.0),(-0.4,1.0)$ | 3.83 | $(-1.0,1.4),(-0.4,1.0)$ | 3.94 |

Table 3: Ablation studies on different time sampling schemes, evaluated by 1-NFE FID.

Figure 8: FID trend on the sampling schemes.

| $p(w)$ | 1-NFE |
|---|---|
| rand., $w=1.5$ | 9.37 |
| rand., $w=2$ | 5.14 |
| const., $w=1.5$ | 6.66 |
| const., $w=2$ | **4.81** |

| EMA rate $\gamma$ | 1-NFE |
|---|---|
| $\gamma=0$ | 10.24 |
| $\gamma=0.9$ | 5.08 |
| $\gamma=0.99$ | **4.90** |
| $\gamma=0.999$ | 6.04 |

| Scaled Param. | 1-NFE | 2-NFE |
|---|---|---|
| yes, $w=2$ | **3.72** | 3.35 |
| no, $w=2$ | 3.82 | **3.27** |
| yes, $w=1.5$ | **6.04** | **4.60** |
| no, $w=1.5$ | 9.32 | 7.02 |

| % t=s | 1-NFE | 2-NFE |
|---|---|---|
| 0 | **3.72** | 3.35 |
| 10% | 3.91 | 3.18 |
| 20% | 3.88 | **2.97** |
| 30% | 3.97 | 3.07 |

(a) Random vs. constant CFG sampling evaluated at example $w$'s.

(b) EMA of pseudo-target $\theta_{\text{sg}}^*$.

(c) With vs. without scaled parameterization.

(d) Prob. for $t = s$ during training.

Table 4: FID ablation on various sampling/parameterization decisions.

the case of random CFG, this velocity can vary widely in magnitude which, if propagated through JVP, can cause wide fluctuation in gradient norm (see left two in Figure 7) and causes training instability. Our method, in comparison, enjoys much smoother gradient norm and $\mathbf{u}(\mathbf{x}_t, t)$ norm, and successfully converges even in the presence of random CFG. We note that random sampling of CFG is not optimal as some CFG scales experience degradation in FID during training, and constant CFG performs better in comparison. We postulate that the under-performance of random CFG is due to limited capacity of the network and the $1/w^2$ factor that downweights high CFG. This phenoemenon is similarly observed in CFG-conditioned FM training (see Appendix F.3) and we leave any improved design to future work.

**No schedules and loss modification.** We do not rely on training curriculum such as warmup schedules in sCT. For each CFG scale, we use the default CFG velocity for all $t, s$, while MeanFlow relies on additional hyperparameters to turn on CFG only when $t$ is within a predetermined range. We also strictly adhere to the simple $\mathcal{L}_2$ loss without any adaptive weighting as proposed by MeanFlow. We believe the simplicity in our design allows for more scalability.

## 7.3 ABLATION STUDIES

We ablate various implementation decisions and discuss insights from different parameter choices. Results are presented with XL/2 architecture trained for 200K steps with batch size 1024.

**Time sampling.** Similar to Flow Matching, different time sampling schemes can greatly affect performance. We explore 3 different kinds of sampling schemes.

- **Truncated sampling** (trunc). Let $(\mu_t,\sigma_t),(\mu_s,\sigma_s)$ denote $t$ being sampled from logit-normal distribution with mean and standard deviation $(\mu_t,\sigma_t)$ and $s$ beinsg sampled from truncated logit-normal distribution with parameters $(\mu_s,\sigma_s)$ such that $s \leq t$.
- **Clamped independent sampling** (clamp). Let $(\mu_t,\sigma_t),(\mu_s,\sigma_s)$ denote $t$ and $s$ being independently sampled from logit-normal distributions with mean and standard deviation $(\mu_t,\sigma_t)$ and $(\mu_s,\sigma_s)$, and set $s = t$ if $s > t$.
- **Truncated gap sampling** (gap). Let $(\mu_g,\sigma_g),(\mu_s,\sigma_s)$ denote the gap $g = t - s$ being sampled from logit-normal distribution with mean and standard deviation $(\mu_g,\sigma_g)$, and $s$ sampled from logit-normal with parameters $(\mu_s,\sigma_s)$ truncated at $1 - g$. Then set $t = s + g$.

In Table 3 we show comparison within each sampling scheme and conclude that better results are obtained when $t$ is biased towards 1 and $s$ biased towards 0 for the model to learn taking longer strides. However, biasing too much, e.g. $\mu_t = 2.0, \sigma_t = 2.0$, leads to worse results. For gap, sampling $t - s$ with lower mean is preferrable to higher mean. In Figure 8, we also observe trunc's performance degrades and clamp plateaus faster than gap. Therefore gap wins over longer training horizons.

All above sampling schemes follow the naïve joint sampling of $(s, t)$. We lastly explore separate time distribution for the FM loss term (see Section 4). We follow gap-sampler and denote the sampler gap$\star$ with parameters $(\mu_g, \sigma_g), (\mu_s, \sigma_s)$ where $(s, t)$ is jointly sampled for the first loss term and $(\mu_s, \sigma_s)$ is used to construct a new logit-normal distribution to independently sample $s'$ for the FM loss. We find in Figure 9 that gap$\star$ generally performs better in 1-NFE FID.

| | FID |
|---|---|
| gap $(-0.8, 1.0), (-0.4, 1.0)$ | 3.44 |
| gap* $(-0.8, 1.0), (-0.4, 1.0)$ | **3.36** |

Figure 9: Sampler comparisons.

**CFG sampling.** As described in the previous section, due to limited capacity of the model, we observe tradeoff in performance when CFG is randomly sampled during training. This is reflected in Table 4a. We note that constant CFG always outperforms random CFG, and for constant CFG sampling we find $w = 2$ converging faster than the default $w = 1.5$ for Flow Matching.

**EMA target rate $\gamma$.** The target EMA weight $\theta^*$ plays a significant role in accelerating convergence of the model. Shown in Table 4b, non-EMA target, *i.e.* $\gamma = 0$, noticeably lags behind $\gamma > 0$ alternatives. However, too large of a $\gamma$, *e.g.* 0.9999, also causes instability because of the overly slow target update. A sweet spot exists around $\gamma = 0.99$ which we use as default. Besides attribute its success to variance reduction because EMA's slower weight update implies much lower optimization noise. In addition, EMA is commonly used to evaluate diffusion models for its quality boost (Song et al., 2020), so being the optimization target also provides better learning signal to the model.

**Scaled parameterization.** In Table 4c, we find scaled parameterization is generally beneficial, but its benefit may vary depending on training/data settings. We therefore suggest testing different choices for different settings for best performance.

**Probability for $t = s$.** Inspired by MeanFlow (Geng et al., 2025), we investigate whether setting $t = s$ (when it reduces to pure FM training) is helpful for overall performance. We find that $> 0\%$ actually degrades 1-NFE performance while it marginally improves 2-NFE performance. This tradeoff persists throughout the training but we observe diminishing return as training goes on. Therefore, we do not find this practice helpful in general and leave it out of our design space in general.

### 7.4 MEMORY AND RUNTIME ANALYSIS

We analyze per-step runtime and per-GPU memory consumption (averaged over 10 training steps without counting EMA update cost) without any performance optimization (*e.g.* torch.compile). Shown on the right is a comparison with Mean-Flow using 256 batch size on 8-GPU H100 cluster on ImageNet-256×256. Since JVP with Flash Attention is not officially sup-

| | Runtime (s) | Memory (GB) |
|---|---|---|
| MeanFlow (w/ naïve SDPA) | - | OOM |
| MeanFlow (w/ our kernel) | 0.81 | **46.73** |
| TVM (w/ improved DiT) | 0.95 | 71.44 |
| TVM (w/ naïve DiT) | 0.86 | 59.53 |
| TVM (w/ improved DiT, detach JVP) | **0.69** | 55.71 |

Figure 10: Per-step time and per-GPU memory study.

ported by PyTorch, the simplest way to implement MeanFlow is to use naïve SDPA, which runs out of memory. MeanFlow with our kernel does not run OOM. TVM with Lipschitz control (Section 4) experiences higher runtime and memory mostly due to architectural change, since TVM with naïve DiT is only marginally more expensive than MeanFlow with naïve DiT. We note that much of the additional compute can be compiled away via PyTorch. Additionally, if step time is a concern, we can simply detach the JVP which biases learning gradient but dramatically reduces runtime. We leave further efficiency optimization to future work.

## 8 CONCLUSION

We present Terminal Velocity Matching, a framework for training one/few-step generative model from scratch. Different from prior works, we match the terminal velocity of a flow trajectory instead of the initial velocity, and we show our objective can explicitly upper bound 2-Wasserstein distance up to a constant. Our proposed objective is conceptually simple and easy to implement, and our theory sheds light on flaws of current diffusion transformers for their lack of Lipschitz continuity. TVM achieves state-of-the-art one-step result for a model trained from scratch and surpasses baseline diffusion models with only 4 NFEs. We hope it can provide new insights into making scalable and performant one/few-step generative paradigms to come.

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

# A    THEOREMS AND DERIVATIONS

## A.1    GENERAL NETWORK PARAMETERIZATION

In general, we can parameterize our net displacement as

$$\mathbf{f}_\theta(\mathbf{x}_t, t, s) = \gamma(t, s)\mathbf{F}_\theta(\mathbf{x}_t, t, s) \tag{20}$$

for some $\gamma(t, s)$ that satisfies $\gamma(t, t) = 0$ for boundary condition. And for the velocity condition, we let

$$\mathbf{u}_\theta(\mathbf{x}_t, t) := \frac{\mathrm{d}}{\mathrm{d}s}\mathbf{f}_\theta(\mathbf{x}_t, t, s)\Big|_{s=t} = \bar{\gamma}(t)\mathbf{F}_\theta(\mathbf{x}_t, t, t) \tag{21}$$

where $\bar{\gamma}(t) = \partial_s\gamma(t, s)|_{s=t}$.

We derive $\frac{\mathrm{d}}{\mathrm{d}s}\mathbf{f}_\theta(\mathbf{x}_t, t, s)|_{s=t}$ below for clarity.

$$\frac{\mathrm{d}}{\mathrm{d}s}\mathbf{f}_\theta(\mathbf{x}_t, t, s)\Big|_{s=t} = \partial_s\gamma(t, s)\mathbf{F}_\theta(\mathbf{x}_t, t, s) + \gamma(t, s)\partial_s\mathbf{F}_\theta(\mathbf{x}_t, t, s)\Big|_{s=t} \tag{22}$$

$$= \partial_s\gamma(t, s)|_{s=t}\mathbf{F}_\theta(\mathbf{x}_t, t, t) + \gamma(t, t)\left[\partial_s\mathbf{F}_\theta(\mathbf{x}_t, t, s)\Big|_{s=t}\right] \tag{23}$$

$$= \partial_s\gamma(t, s)|_{s=t}\mathbf{F}_\theta(\mathbf{x}_t, t, t) \tag{24}$$

$$= \bar{\gamma}(t)\mathbf{F}_\theta(\mathbf{x}_t, t, t) \tag{25}$$

## A.2    TERMINAL VELOCITY ERROR UPPER BOUNDS DISPLACEMENT ERROR

**Lemma 1.** *Under mild regularity assumptions, the following inequality holds,*

$$\mathcal{L}_{displ}^t(\theta) \leq \int_0^t \mathbb{E}_{\mathbf{x}_t}\left[\left\|\frac{\mathrm{d}}{\mathrm{d}s}\mathbf{f}_\theta(\mathbf{x}_t, t, s) - \mathbf{u}(\psi(\mathbf{x}_t, t, s), s)\right\|_2^2\right]\mathrm{d}s \tag{26}$$

*where $p_t(\mathbf{x}_t)$ is marginal distributions for initial points $\mathbf{x}_t$.*

*Proof.* We assume both displacement maps are Riemann-integrable, then

$$\mathcal{L}_{\mathrm{displ}}^t(\theta) = \mathbb{E}_{\mathbf{x}_t \sim p_t(\mathbf{x}_t)}\left[\left\|\mathbf{f}_\theta(\mathbf{x}_t, t, 0) - \int_t^0 \mathbf{u}(\mathbf{x}_s, s)\mathrm{d}s\right\|_2^2\right] \tag{27}$$

$$= \mathbb{E}_{\mathbf{x}_t \sim p_t(\mathbf{x}_t)}\left[\left\|\int_0^t \frac{\mathrm{d}}{\mathrm{d}s}\mathbf{f}_\theta(\mathbf{x}_t, t, s)\mathrm{d}s - \int_0^t \mathbf{u}(\psi(\mathbf{x}_t, t, s), s)\mathrm{d}s\right\|_2^2\right] \tag{28}$$

$$\overset{(*)}{\leq} \int_0^t \mathbb{E}_{\mathbf{x}_t \sim p_t(\mathbf{x}_t)}\left[\left\|\frac{\mathrm{d}}{\mathrm{d}s}\mathbf{f}_\theta(\mathbf{x}_t, t, s) - \mathbf{u}(\psi(\mathbf{x}_t, t, s), s)\right\|_2^2\right]\mathrm{d}s \tag{29}$$

where $(*)$ uses triangle inequality and regularity assumption. $\qquad\square$

## A.3    TERMINAL VELOCITY ERROR REDUCES TO FM

Consider the terminal velocity error for each time $s$ as

$$\mathbb{E}_{\mathbf{x}_t}\left[\left\|\frac{\mathrm{d}}{\mathrm{d}s}\mathbf{f}_\theta(\mathbf{x}_t, t, s) - \mathbf{u}(\psi(\mathbf{x}_t, t, s), s)\right\|_2^2\right] \tag{30}$$

Expand the inner term

$$\frac{\mathrm{d}}{\mathrm{d}s}\mathbf{f}_\theta(\mathbf{x}_t, t, s) = \mathbf{F}_\theta(\mathbf{x}_t, t, s) + (s - t)\partial_s\mathbf{F}_\theta(\mathbf{x}_t, t, s) \tag{31}$$

and for the inner norm term its limit exists as $t \to s$:

$$\lim_{t \to s} \left[ \frac{\mathrm{d}}{\mathrm{d}s} \mathbf{f}_\theta(\mathbf{x}_t, t, s) - \mathbf{u}(\psi(\mathbf{x}_t, t, s), s) \right] \tag{32}$$

$$= \lim_{t \to s} \left[ \mathbf{F}_\theta(\mathbf{x}_t, t, s) + (s - t)\partial_s \mathbf{F}_\theta(\mathbf{x}_t, t, s) - \mathbf{u}(\psi(\mathbf{x}_t, t, s), s) \right] \tag{33}$$

$$= \mathbf{F}_\theta(\mathbf{x}_s, s, s) - \mathbf{u}(\mathbf{x}_s, s) \tag{34}$$

Thus, the limit of its expected $\mathcal{L}_2$-norm exists (assuming this norm is bounded) and is equal to $\mathcal{L}_2$-norm of its limit, which is

$$\mathbb{E}_{\mathbf{x}_s} \left[ \| \mathbf{F}_\theta(\mathbf{x}_s, s, s) - \mathbf{u}(\mathbf{x}_s, s) \|_2^2 \right] \tag{35}$$

and this is the original FM loss, which is equivalent (up to a constant) to conditional Flow Matching loss used in practice in Eq. (1).

## A.4 MAIN THEOREM

**Theorem 1** (Connection to the 2-Wasserstein distance). *Given $t \in [0, 1]$, let $\mathbf{f}_{t \to 0}^\theta \# p_t(\mathbf{x}_t)$ be the distribution pushforward from $p_t(\mathbf{x}_t)$ via $\mathbf{f}_\theta(\mathbf{x}_t, t, 0)$, and assume $\mathbf{u}_\theta(\cdot, s)$ is Lipschitz-continuous for all $s \in [0, t]$ with Lipschitz constants $L(s)$, with additional mild regularity conditions,*

$$W_2^2(\mathbf{f}_{t \to 0}^\theta \# p_t, p_0) \leq \int_0^t \lambda[L](s) \mathcal{L}_{\mathrm{TVM}}^{t,s}(\theta) \mathrm{d}s + C, \tag{10}$$

*where $W_2(\cdot, \cdot)$ is 2-Wasserstein distance, $\lambda[\cdot]$ is a functional of $L(\cdot)$, and $C$ is a non-optimizable constant.*

*Proof.* Note that the ground-truth flow map $\psi$ is invertible and that $\psi(\psi(\mathbf{x}_t, t, 0), 0, t) = \mathbf{x}_t$ and $\psi(\psi(\mathbf{x}_0, 0, t), t, 0) = \mathbf{x}_0$.

$$W_2^2(\mathbf{f}_{t \to 0}^\theta \# p_t, p_0) \overset{(i)}{\leq} \int p_0(\mathbf{x}_0) \| \mathbf{f}_\theta(\psi(\mathbf{x}_0, 0, t), t, 0) - \mathbf{x}_0 \|_2^2 \mathrm{d}\mathbf{x}_0 \tag{36}$$

$$= \int p_0(\mathbf{x}_0) \| \mathbf{x}_t + \mathbf{f}_\theta(\psi(\mathbf{x}_0, 0, t), t, 0) - \psi(\psi(\mathbf{x}_0, 0, t), t, 0) \|_2^2 \mathrm{d}\mathbf{x}_0 \tag{37}$$

$$= \int p_t(\mathbf{x}_t) \| \mathbf{x}_t + \mathbf{f}_\theta(\mathbf{x}_t, t, 0) - \psi(\mathbf{x}_t, t, 0) \|_2^2 \mathrm{d}\mathbf{x}_t \tag{38}$$

$$= \int p_t(\mathbf{x}_t) \left\| \int_t^0 \frac{\mathrm{d}}{\mathrm{d}s} \mathbf{f}_\theta(\mathbf{x}_t, t, s) \mathrm{d}s - \int_t^0 \mathbf{u}(\mathbf{x}_s, s) \mathrm{d}s \right\|_2^2 \mathrm{d}\mathbf{x}_t \tag{39}$$

$$\overset{(ii)}{\leq} \int p_t(\mathbf{x}_t) \int_0^t \underbrace{\left\| \frac{\mathrm{d}}{\mathrm{d}s} \mathbf{f}_\theta(\mathbf{x}_t, t, s) - \mathbf{u}(\psi(\mathbf{x}_t, t, s), s) \right\|_2^2}_{\varepsilon(\mathbf{x}_t, t, s)} \mathrm{d}s \mathrm{d}\mathbf{x}_t \tag{40}$$

where $(i)$ is due to Wasserstein distance being the infimum of all couplings, and we choose a particular coupling of the two distribution by inverting $\mathbf{x}_0$ with $\psi$ and remapping with respective flow maps. And $(ii)$ is due to Lemma 1. Now, we inspect $\varepsilon(\mathbf{x}_t, t, s)$ specifically by noticing that

$$\varepsilon(\mathbf{x}_t, t, s)$$

$$= \| \frac{\mathrm{d}}{\mathrm{d}s} \mathbf{f}_\theta(\mathbf{x}_t, t, s) - \mathbf{u}(\psi(\mathbf{x}_t, t, s), s) + \mathbf{u}_\theta(\psi(\mathbf{x}_t, t, s), s) - \mathbf{u}_\theta(\psi(\mathbf{x}_t, t, s), s)$$
$$\quad + \mathbf{u}_\theta(\mathbf{f}_\theta(\mathbf{x}_t, t, s), s) - \mathbf{u}_\theta(\mathbf{f}_\theta(\mathbf{x}_t, t, s), s) \|^2 \tag{41}$$

$$\overset{(i)}{\leq} \underbrace{\left\| \frac{\mathrm{d}}{\mathrm{d}s} \mathbf{f}_\theta(\mathbf{x}_t, t, s) - \mathbf{u}_\theta(\mathbf{f}_\theta(\mathbf{x}_t, t, s), s) \right\|_2^2 + \| \mathbf{u}_\theta(\psi(\mathbf{x}_t, t, s), s) - \mathbf{u}(\psi(\mathbf{x}_t, t, s), s) \|_2^2}_{\delta(\mathbf{x}_t, t, s)}$$

$$+ \| \mathbf{u}_\theta(\mathbf{f}_\theta(\mathbf{x}_t, t, s), s) - \mathbf{u}_\theta(\psi(\mathbf{x}_t, t, s), s) \|_2^2 \tag{42}$$

$$\overset{(ii)}{\leq} \delta(\mathbf{x}_t, t, s) + L(s) \int_s^t \underbrace{\left\| \frac{\mathrm{d}}{\mathrm{d}s} \mathbf{f}_\theta(\mathbf{x}_t, t, u) - \mathbf{u}(\psi(\mathbf{x}_t, t, u), u) \right\|_2^2}_{\varepsilon(\mathbf{x}_t, t, u)} \mathrm{d}u \tag{43}$$

where $(i)$ is due to triangle inequality and $(ii)$ is due to Lipschitz-continuous assumption. We further notice that right-hand-side contains a term that is the integral of the left-hand-side. For simplicity, we hold $\mathbf{x}_t$ and $t$ constant and let

$$y(s) = \int_s^t \varepsilon(\mathbf{x}_t, t, u)\mathrm{d}u \,, \qquad \dot{y}(s) = -\varepsilon(\mathbf{x}_t, t, s)$$

and we arrive at the following inequality,

$$-\dot{y}(s) \leq \delta(\mathbf{x}_t, t, s) + L(s)y(s) \tag{44}$$

$$-\delta(\mathbf{x}_t, t, s) \leq \dot{y}(s) + L(s)y(s) \tag{45}$$

$$-e^{\int_t^r L(u)\mathrm{d}u}\delta(\mathbf{x}_t, t, r) \leq \frac{\mathrm{d}}{\mathrm{d}r}\left(e^{\int_t^r L(u)\mathrm{d}u}y(r)\right) \tag{46}$$

$$-\int_s^t e^{\int_t^r L(u)\mathrm{d}u}\delta(\mathbf{x}_t, t, r)\mathrm{d}r \leq \left[e^{\int_t^r L(u)\mathrm{d}u}y(r)\right]_s^t \tag{47}$$

$$-\int_s^t e^{\int_t^r L(u)\mathrm{d}u}\delta(\mathbf{x}_t, t, r)\mathrm{d}r \leq \overset{0}{\cancel{y(t)}} - e^{\int_t^s L(u)\mathrm{d}u}y(s) \tag{48}$$

$$e^{\int_t^s L(u)\mathrm{d}u}y(s) \leq \int_s^t e^{\int_t^r L(u)\mathrm{d}u}\delta(\mathbf{x}_t, t, r)\mathrm{d}r \tag{49}$$

$$y(s) \leq \int_s^t e^{\int_t^r L(u)\mathrm{d}u - \int_t^s L(u)\mathrm{d}u}\delta(\mathbf{x}_t, t, r)\mathrm{d}r \tag{50}$$

$$y(s) \leq \int_s^t e^{\int_s^r L(u)\mathrm{d}u}\delta(\mathbf{x}_t, t, r)\mathrm{d}r \tag{51}$$

Therefore, setting $s = 0$ we have

$$\int_0^t \varepsilon(\mathbf{x}_t, t, u)\mathrm{d}u \leq \int_0^t \underbrace{e^{\int_0^r L(u)\mathrm{d}u}}_{\lambda[L](r)}\delta(\mathbf{x}_t, t, u)\mathrm{d}u \tag{52}$$

where the left-hand side is the inner term of Eq. (40). Then,

$$\text{Eq. (40)} \leq \int p_t(\mathbf{x}_t)\int_0^t \lambda[L](s) \cdot \delta(\mathbf{x}_t, t, s)\mathrm{d}s\mathrm{d}\mathbf{x}_t \tag{53}$$

$$= \int_0^t \lambda[L](s) \cdot \mathbb{E}_{\mathbf{x}_t}\left[\|\mathbf{f}_\theta(\mathbf{x}_t, t, s) - \mathbf{u}_\theta(\mathbf{f}_\theta(\mathbf{x}_t, t, s), s)\|_2^2\right.$$
$$\left. + \|\mathbf{u}_\theta(\psi(\mathbf{x}_t, t, s), s) - \mathbf{u}(\psi(\mathbf{x}_t, t, s), s)\|_2^2\right]\mathrm{d}s \tag{54}$$

$$= \int_0^t \lambda[L](s) \cdot \left[\mathbb{E}_{\mathbf{x}_t}\left[\|\mathbf{f}_\theta(\mathbf{x}_t, t, s) - \mathbf{u}_\theta(\mathbf{f}_\theta(\mathbf{x}_t, t, s), s)\|_2^2\right.\right.$$
$$\left.\left. + \mathbb{E}_{\mathbf{x}_t}\left[\|\mathbf{u}_\theta(\psi(\mathbf{x}_t, t, s), s) - \mathbf{u}(\psi(\mathbf{x}_t, t, s), s)\|_2^2\right]\right]\mathrm{d}s \tag{55}$$

$$= \int_0^t \lambda[L](s) \cdot \left[\mathbb{E}_{\mathbf{x}_t}\left[\|\mathbf{f}_\theta(\mathbf{x}_t, t, s) - \mathbf{u}_\theta(\mathbf{f}_\theta(\mathbf{x}_t, t, s), s)\|_2^2\right.\right.$$
$$\left.\left. + \underbrace{\mathbb{E}_{\mathbf{x}_s}\left[\|\mathbf{u}_\theta(\mathbf{x}_s, s) - \mathbf{u}(\mathbf{x}_s, s)\|_2^2\right]}_{(a)}\right]\mathrm{d}s \tag{56}$$

where $(a)$ can be rewritten as

$$(a) = \mathbb{E}_{\mathbf{x}_s, \mathbf{v}_s}\left[\|\mathbf{u}_\theta(\mathbf{x}_s, s) - \mathbf{v}_s\|_2^2\right] + \tilde{C} \tag{57}$$

where $\tilde{C}$ is some non-optimizable constant (Lipman et al., 2022). This is also a classical result connecting score matching and denoising score matching (Vincent, 2011).

Now, after substitution, we notice that our bound in Eq. (56) becomes

$$\int_0^t \lambda[L](s)\mathcal{L}_{\text{TVM}}^{t,s}(\theta)\mathrm{d}s + C \tag{58}$$

where $C$ is some other constant, which completes the proof. $\square$

### A.5 Reduction to Flow Matching

When $t = s$, we show that $\mathcal{L}_{\text{TVM}}(\theta)$ reduces to Flow Matching loss.

$$\mathcal{L}_{\text{TVM}}^{t,t}(\theta) = \mathbb{E}_{\mathbf{x}_t,\mathbf{x}_s,\mathbf{v}_s} \left[ \left\| \frac{\mathrm{d}}{\mathrm{d}s}\mathbf{f}_\theta(\mathbf{x}_t,t,s) - \mathbf{u}_\theta(\mathbf{f}_\theta(\mathbf{x}_t,t,s),s) \right\|_2^2 + \left\| \mathbf{u}_\theta(\mathbf{x}_s,s) - \mathbf{v}_s \right\|^2 \right] \Bigg|_{s=t} \tag{59}$$

$$= \mathbb{E}_{\mathbf{x}_t,\mathbf{v}_t} \left[ \cancel{\|\mathbf{u}_\theta(\mathbf{x}_t,t) - \mathbf{u}_\theta(\mathbf{x}_t,t)\|_2^2} + \left\| \mathbf{u}_\theta(\mathbf{x}_t,t) - \mathbf{v}_t \right\|^2 \right] \tag{60}$$

### A.6 Derivation for class-conditional training target

In Eq. (14), we introduced the CFG training target as

$$w\mathbf{v}_t + (1-w)\mathbf{u}_{\theta_{\text{sg}}^*}^1(\mathbf{x}_s,s,\varnothing)$$

We derive below that the minimizer of Eq. (14) is the CFG velocity $w\mathbf{u}(\mathbf{x}_s,s,c) + (1-w)\mathbf{u}(\mathbf{x}_s,s)$.

*Proof.* Consider the training objective (without weighting for simplicity)

$$\mathbb{E}_{\mathbf{x}_s,\mathbf{v}_s,s,c,w} \left[ \left\| \mathbf{u}_\theta^w(\mathbf{x}_s,s,c) - \left[ w\mathbf{v}_t + (1-w)\mathbf{u}_{\theta_{\text{sg}}}^1(\mathbf{x}_s,s,\varnothing)) \right] \right\|_2^2 \right] \tag{61}$$

when $c = \varnothing, w = 1$, then it reduces to

$$\mathbb{E}_{\mathbf{x}_s,\mathbf{v}_s,s} \left[ \left\| \mathbf{u}_\theta^1(\mathbf{x}_s,s,\varnothing) - \mathbf{v}_t \right\|_2^2 \right] \tag{62}$$

with the minimizer $\theta_{\min}$ satisfying $\mathbf{u}_{\theta_{\min}}^1(\mathbf{x}_s,s,\varnothing) = \mathbf{u}(\mathbf{x}_s,s)$.

At minimum of the loss for other $w$ and $c$, it must satisfy

$$\mathbf{u}_{\theta_{\min}}^w(\mathbf{x}_s,s,c) = \mathbb{E}_{\mathbf{v}_s} \left[ w\mathbf{v}_s + (1-w)\mathbf{u}_{\theta_{\min}}^1(\mathbf{x}_s,s,\varnothing) \mid \mathbf{x}_s,s,c,w \right] \tag{63}$$

$$= w\mathbb{E}_{\mathbf{v}_s} \left[ \mathbf{v}_s \mid \mathbf{x}_s,s,c \right] + (1-w)\mathbf{u}_{\theta_{\min}}^1(\mathbf{x}_s,s,\varnothing) \tag{64}$$

$$= w\mathbf{u}(\mathbf{x}_s,s,c) + (1-w)\mathbf{u}(\mathbf{x}_s,s) \tag{65}$$

$\square$

## B Additional Details on Practical Challenges

### B.1 Lipschitzness of RMSNorm

Recall the definition of RMSNorm, for input $x \in \mathbb{R}^d$ and a small constant $\epsilon > 0$

$$\text{RMSNorm}(x) = \frac{x}{\text{RMS}(x)}, \quad \text{where} \quad \text{RMS}(x) = \sqrt{\frac{1}{d}\sum_{i=1}^d x_i + \epsilon} \tag{66}$$

And its Jacobian can be calculated as

$$\frac{\mathrm{d}}{\mathrm{d}x_j}\text{RMSNorm}(x_i) = \frac{\mathrm{d}}{\mathrm{d}x_j}\left( \frac{x_i}{\text{RMS}(x)} \right) \tag{67}$$

$$= \frac{\delta_{ij}\text{RMS}(x) - x_i x_j/\text{RMS}(x)/d}{\text{RMS}(x)^2} \tag{68}$$

$$= \frac{\delta_{ij}}{\text{RMS}(x)} - \frac{x_i x_j}{d \cdot \text{RMS}(x)^3} \tag{69}$$

```
def get_f_and_dfds(net, xt, t, s, c, w):
    def model_wrapper(x_, t_, s_):  # we use t-s for second time condition
        return net(x_, t_, (t_ - s_), c, w)
    F, dFds = torch.func.jvp(model_wrapper, (xt, t, s), (0, 0, 1))
    f_ts = xt + (s - t) * F
    dfds = (F + (s - t) * dFds)
    return f_ts, dfds
```

Figure 11: PyTorch-style JVP code.

Since matrix norm (largest singular value) $\sigma(A)$ of matrix $A$ is upper bounded by its Frobenius norm, and $\text{RMS}(x) \geq \epsilon$, we have each element $\frac{\text{d}}{\text{d}x_j}\text{RMSNorm}(x_i)$ in the Jacobian matrix bounded via

$$\left| \frac{\text{d}}{\text{d}x_j}\text{RMSNorm}(x_i) \right|^2 \leq \left| \frac{\delta_{ij}}{\text{RMS}(x)} \right|^2 + \left| \frac{x_i x_j}{d \cdot \text{RMS}(x)^3} \right|^2 \tag{70}$$

$$= \left| \frac{\delta_{ij}}{\text{RMS}(x)} \right|^2 + \left( \frac{x_i/\sqrt{d}}{\text{RMS}(x)} \right)^2 \cdot \left( \frac{x_j/\sqrt{d}}{\text{RMS}(x)} \right)^2 \cdot \frac{1}{\text{RMS}(x)}^2 \tag{71}$$

$$\leq \frac{1}{\epsilon} + \frac{1}{\epsilon} \tag{72}$$

$$= \frac{2}{\epsilon} \tag{73}$$

Therefore, the Frobenius norm is bounded and hence the matrix norm.

### B.2 FULL DESCRIPTION OF NORMALIZATION OF MODULATION

Note that there are 6 modulation parameters in total for each DiT layer, denoted as

$$a_1(t), b_1(t), c_1(t), a_2(t), b_2(t), c_2(t) = \text{split}(\text{AdaLN\_Modulation}(t), 6) \tag{74}$$

and we pass each of the above parameters through $\text{RMSNorm}^-(\cdot)$ to obtain

$$a_1^-(t), b_1^-(t), c_1^-(t), a_2^-(t), b_2^-(t), c_2^-(t)$$

(which can be done in parallel) and the new normalized DiT layer is

$$x = x + c_1^-(t) * \text{ATTN}(\text{RMSNorm}^-(x) * a_1^-(t) + b_1^-(t))$$
$$x = x + c_2^-(t) * \text{MLP}(\text{RMSNorm}^-(x) * a_2^-(t) + b_2^-(t))$$

## C FLASH ATTENTION JVP WITH BACKWARD PASS

In transformer models, scaled dot-product attention (SDPA) is often among the most, if not the most, computationally expensive operations. The cost stems not only from its high FLOP requirements – $O(MN)$ in general, and $O(N^2)$ in the case of self-attention – but also from the quadratic memory footprint of the query–key matrix multiplication.

Computing the Jacobian-Vector Product (JVP) of SDPA is even more demanding, typically requiring about three times the cost of the standard forward pass. Flash attention (Dao et al., 2022) fuses the matrix multiplication with an online softmax operation (Milakov & Gimelshein, 2018), thereby eliminating the need to store the intermediate $QK^\top$ matrix in GPU memory. Subsequent work has shown that JVP SDPA can also be implemented in a FlashAttention-style manner, where both, primal SDPA and JVP SDPA are computed jointly to avoid redundant computation (Lu & Song, 2024).

Building on these ideas, we implement efficient JVP SDPA forward and backward kernels in Triton. We first take inspiration from open-source implementations without backward support[4]. And the

---

[4]https://github.com/Ryu1845/min-sCM/blob/main/standalone_multihead_jvp_test.py

additional backward pass through the standard ("primal") SDPA is handled independently using the open-source implementation from (Dao et al., 2022). To obtain full gradients with respect to $Q$, $K$, and $V$, we combine the input gradients from both backward passes.

Similar to standard SDPA, the JVP backward pass can leverage online softmax to avoid storing large intermediate matrices in GPU memory. However, the increased complexity of JVP SDPA requires additional optimizations to run efficiently on GPUs. Most notably, we found it crucial to split the backward computation into multiple smaller kernels to reduce register spills caused by the large number of intermediate tensors.

**Background.** Recall the attention operation as

$$\text{ATTN}(Q, K, V) = V \cdot \text{softmax}\left(\frac{QK^T}{\sqrt{d_k}}\right) \tag{75}$$

and let the query, key, and value blocks be denoted by $Q \in \mathbb{R}^{M \times d}$, $K \in \mathbb{R}^{N \times d}$ and $V \in \mathbb{R}^{N \times d}$. The tangent inputs are denoted as $\dot{Q}$, $\dot{K}$, $\dot{V}$. We use $\alpha = \frac{1}{\sqrt{d_k}}$ as the softmax scaling factor, and $\ell_i$ denotes the log-sum-exponential normalization for the $i$-th row of the attention scores, a short form for combining the softmax stabilization factor and the normalization.

## C.1 MULTI-STEP BACKWARD PASS

For best performance, we decided to split up the backward pass into multiple smaller operations with shared paths through the graph. Furthermore, the gradients $dQ$ and $d\dot{Q}$ are computed in row-parallel order, while $dK$, $d\dot{K}$, $dV$ and $d\dot{V}$ are processed in column-parallel order. In our tests, redundant, but coalesced computation of the large parts of the backward pass greatly outperformed a single, fused kernel relying on atomic operations.

We split the operation into 6 steps: 1) preprocess shared intermediates, 2) process $d\dot{K}$ and first part of $dK$, 3) process $d\dot{Q}$ and first part of $dQ$, 4) process second part of $dK$, 5) process second part of $dQ$, 6) process $d\dot{V}$ and $dV$.

**Step 1: Preprocess shared intermediates row-parallel.** In the first step, we preprocess two intermediate sums $\Sigma_1 \in \mathbb{R}^M$ and $\Sigma_2 \in \mathbb{R}^M$ used in steps 2-5.

$$\Sigma_{1,i} = \sum_j P_{ij}\left(d\dot{O}V^\top\right)_{ij} \tag{76}$$

$$\Sigma_{2,i} = \sum_j P_{ij}\left(\left(d\dot{O}\dot{V}^\top\right)_{ij} + \left(d\dot{O}V^\top\right)_{ij} N_{ij}\right) \tag{77}$$

where

$$P_{ij} = \exp\left(\alpha S_{ij} - \ell_i\right), \quad S_{ij} = \sum_{r=1}^{d_k} Q_{ir} K_{jr}, \quad N_{ij} = \alpha \dot{S}_{ij} - \frac{\mu_i}{l_i} \tag{78}$$

amd

$$\dot{S}_{ij} = \sum_{r=1}^{d_k} \left(\dot{Q}_{ir} K_{jr} + Q_{ir} \dot{K}_{jr}\right), \quad \mu_i = \sum_j P_{ij}\left(\alpha \dot{S}_{ij}\right) \tag{79}$$

**Step 2: process $d\dot{K}$ and $dK_1$ column-parallel.**

$$(dK_1)_{j,:} = \alpha \sum_i \left[\left(\left(d\dot{O}V^\top\right)_{ij} - \Sigma_{1,i}\right) P_{ij}\right] \dot{Q}_{i,:} \tag{80}$$

$$(d\dot{K})_{j,:} = \alpha \sum_i \left[\left(\left(d\dot{O}V^\top\right)_{ij} - \Sigma_{1,i}\right) P_{ij}\right] Q_{i,:} \tag{81}$$

**Step 3: Process $d\dot{Q}$ and $dQ_1$ row-parallel.**

$$(dQ_1)_{i,:} = \alpha \sum_j \left[ \left( \left( d\dot{O}V^\top \right)_{ij} - \Sigma_{1,i} \right) P_{ij} \right] \dot{K}_{j,:} \tag{82}$$

$$(d\dot{Q})_{i,:} = \alpha \sum_j \left[ \left( \left( d\dot{O}V^\top \right)_{ij} - \Sigma_{1,i} \right) P_{ij} \right] K_{j,:} \tag{83}$$

**Step 4: Process $dK$ column-parallel.**

$$
\begin{aligned}
(dK)_{j,:} = (dK_1)_{j,:} + \alpha \sum_i \Bigg\{ &\left[ \alpha \left( -\Sigma_{1,i} \right) \dot{S}_{ij} + \Sigma_{1,i} \frac{\mu_i}{l_i} \right] P_{ij} \\
&+ \left[ \left( d\dot{O}\dot{V}^\top \right)_{ij} + \left( d\dot{O}V^\top \right)_{ij} \left( \alpha \dot{S}_{ij} - \frac{\mu_i}{l_i} \right) - \Sigma_{2,i} \right] P_{ij} \Bigg\} Q_{i,}
\end{aligned}
\tag{84}
$$

**Step 5: Process $dQ$ row-parallel.**

$$
\begin{aligned}
(dQ)_{i,:} = (dQ_1)_{i,:} + \alpha \sum_j \Bigg\{ &\left[ \alpha \left( -\Sigma_{1,i} \right) \dot{S}_{ij} + \Sigma_{1,i} \frac{\mu_i}{l_i} \right] P_{ij} \\
&+ \left[ \left( d\dot{O}\dot{V}^\top \right)_{ij} + \left( d\dot{O}V^\top \right)_{ij} \left( \alpha \dot{S}_{ij} - \frac{\mu_i}{l_i} \right) - \Sigma_{2,i} \right] P_{ij} \Bigg\} K_{j,:}
\end{aligned}
\tag{85}
$$

**Step 6: Process $dV$ and $d\dot{V}$ column-parallel.**

$$(d\dot{V})_{j,:} = \sum_i P_{ij} (d\dot{O})_{i,:} \tag{86}$$

$$(dV)_{j,:} = \sum_j \left[ P_{ij} \left( \alpha \dot{S}_{ij} - \frac{\mu_i}{l_i} \right) \right] (d\dot{O})_{i,:} \tag{87}$$

**Caching softmax statistics.** Like previous flash-attention implementations, we cache softmax statistics from the forward pass to speed up the backward pass, namely the log-sum-exp $\ell$, the sums $l$ and $\mu$ for each row of the output $O$. Thus, the total overhead of the cache is only three values per row of $Q$.

## C.2 EVALUATION

We built a test bench to evaluate latency and peak memory consumption of our flash JVP SDPA kernels on different input shapes using an NVIDIA H100 SXM 80GB. Due to the lack of existing alternatives, we compare against *vanilla* SDPA, i.e. a SDPA written as explicit math operations, which currently is the only way to train transformers in PyTorch with JVP enabled.

As our contribution focuses on the backward pass, we limit the latency and peak memory evaluation to the backward pass of a single SDPA operation, combining both paths through the primal ("normal") and the tangent (JVP) gradients.

**Results.** Shown in Table 5, our implementation achieves a significant reduction in peak memory consumption. Compared to the reference, we save memory not only by reducing the cached variables between forward and backward pass, but more importantly by avoiding to store $N^2$ intermediate attention scores. At the same time, our implementation achieves a speedup of up to 65% compared to the reference.

| H | S | Latency [ms] | | Peak Memory [MB] | |
|---|---|---|---|---|---|
| | | ours | vanilla | ours | vanilla |
| 1 | 128 | 1.31 | 1.51 | 64.69 | 64.80 |
| 1 | 1,024 | 1.38 | 1.54 | 69.52 | 94.02 |
| 1 | 4,096 | 1.96 | 1.53 | 86.06 | 508.1 |
| 1 | 8,192 | 3.98 | 4.33 | 108.1 | 1,816 |
| 1 | 16,384 | 10.06 | 16.11 | 152.3 | 7,024 |
| 1 | 32,768 | 40.24 | 63.85 | 240.5 | 27,808 |
| 24 | 128 | 1.40 | 1.55 | 80.55 | 83.17 |
| 24 | 1,024 | 1.42 | 2.03 | 196.4 | 784.4 |
| 24 | 4,096 | 15.13 | 24.52 | 593.5 | 10,721 |
| 24 | 8,192 | 58.70 | 96.93 | 1,123 | 42,115 |
| 24 | 16,384 | 238.4 | - | 2,182 | - |
| 24 | 32,768 | 958.6 | - | 4,300 | - |

Table 5: Performance comparison of our flash JVP kernels against vanilla SDPA kernels in PyTorch. H and S stand for number of heads in multi head attention and sequence length. Vanilla SDPA ran out of memory on a NVIDIA H100 in the last two tests.

---

**Algorithm 1** TVM Training

---

**Input:** initialized model $\boldsymbol{f}^\theta$, data $p_0(\mathbf{x}_0, c)$ and prior $p_1(\mathbf{x}_1)$, time distribution $p(t, s)$, guidance distribution $p(w)$

Initialize $\theta^* \leftarrow \theta, \theta^{**} \leftarrow \theta$          // $\theta^*, \theta^{**}$ are EMA with rate $\lambda^*, \lambda^{**}$.

**while** model not converged **do**

    Sample $(\mathbf{x}_0, c, \mathbf{x}_1) \sim p_0(\mathbf{x}_0, c)p_1(\mathbf{x}_1)$

    Randomly drop $c$ with prob. 10%

    Sample $(t, s, w) \sim p(t, s)p(w)$          // optionally sample $s' \sim p(s')$ for the second loss term.

    $\mathbf{x}_t \leftarrow (1-t)\mathbf{x}_0 + t\mathbf{x}_1$

    $\mathbf{x}_s \leftarrow (1-s)\mathbf{x}_0 + s\mathbf{x}_1$          // optionally set $\mathbf{x}_{s'} \leftarrow (1-s')\mathbf{x}_0 + s'\mathbf{x}_1$.

    $\mathbf{v}_s \leftarrow \mathbf{x}_1 - \mathbf{x}_0$          // optionally set $\mathbf{v}_{s'} \leftarrow \mathbf{x}_1 - \mathbf{x}_0$.

    $\theta \leftarrow$ optimizer step by minimizing $\hat{\mathcal{L}}_{\text{TVM}}(\theta) = \mathbb{E}_{t,s,w}\left[\hat{\mathcal{L}}_{\text{TVM}}^{t,s,w}(\theta)\right]$          // see Eq. (15)

                                      // optionally use $s'$ and $\mathbf{x}_{s'}$ for the second loss term

    $\theta^* \leftarrow$ EMA update with rate $\lambda^*$

    $\theta^{**} \leftarrow$ EMA update with rate $\lambda^{**}$

**end while**

**Output:** learned model $\boldsymbol{f}^{\theta^{**}}$

---

# D   TRAINING ALGORITHM

We present the training algorithm in Algorithm 1. We additionally show a PyTorch-style pseudo-code in Figure 11 for calculating $\mathbf{f}_\theta(\mathbf{x}_t, t, s)$ and $\frac{\mathrm{d}}{\mathrm{d}s}\mathbf{f}_\theta(\mathbf{x}_t, t, s)$ together with one JVP pass.

# E   RELATION TO OTHER WORKS

## E.1   MEANFLOW

Let $\mathbf{f}_\theta(\mathbf{x}_t, t, s) = (s-t)\mathbf{F}_\theta(\mathbf{x}_t, t, s)$, we inspect

$$\frac{\mathrm{d}}{\mathrm{d}t}\mathbf{f}_\theta(\mathbf{x}_t, t, s) + \mathbf{u}(\mathbf{x}_t, t) \tag{88}$$

$$= -\mathbf{F}_\theta(\mathbf{x}_t, t, s) + (s-t)\frac{\mathrm{d}}{\mathrm{d}t}\mathbf{F}_\theta(\mathbf{x}_t, t, s) + \mathbf{u}(\mathbf{x}_t, t) \tag{89}$$

$$= -\mathbf{F}_\theta(\mathbf{x}_t, t, s) + (s-t)\left[\mathbf{u}(\mathbf{x}_t, t) \cdot \nabla_{\mathbf{x}_t}\mathbf{F}_\theta(\mathbf{x}_t, t, s) + \partial_t\mathbf{F}_\theta(\mathbf{x}_t, t, s)\right] + \mathbf{u}(\mathbf{x}_t, t) \tag{90}$$

|  | ImageNet-256×256 | | ImageNet-512×512 | |
|---|---|---|---|---|
| **Parameterization Setting** | | | | |
| Architecture | DiT-XL/2 | DiT-XL/2 | DiT-XL/2 | DiT-XL/2 |
| Params (M) | 678 | 678 | 678 | 678 |
| $2^{nd}$ time conditioning | $t-s$ | $t-s$ | $t-s$ | $t-s$ |
| Hidden dim | 1152 | 1152 | 1152 | 1152 |
| Number of heads | 18 | 18 | 18 | 18 |
| Main normalization | RMS Norm | RMS Norm | RMS Norm | RMS Norm |
| QK-Norm type | RMS Norm | RMS Norm | RMS Norm | RMS Norm |
| Linear layer init[5] | Spectral | Spectral | Spectral | Spectral |
| Time Embed init[6] | Spectral | $\mathcal{N}(0, 0.02)$ | $\mathcal{N}(0, 0.02)$ | Spectral |
| Training iter | 300K | 300K | 300K | 300K |
| **Training Setting** | | | | |
| Optimizer | AdamW | AdamW | AdamW | AdamW |
| Optimizer $\epsilon$ | $10^{-8}$ | $10^{-8}$ | $10^{-8}$ | $10^{-8}$ |
| $\beta_1$ | 0.9 | 0.9 | 0.9 | 0.9 |
| $\beta_2$ | 0.95 | 0.95 | 0.95 | 0.95 |
| Learning rate | 0.0001 | 0.0001 | 0.0001 | 0.0001 |
| Weight decay | 0 | 0 | 0 | 0 |
| Batch size | 2048 | 2048 | 2048 | 2048 |
| $p(s,t)$ | | gap* $(-0.8, 1.0), (-0.4, 1.0)$ | | |
| Scaled param. | yes | yes | no | yes |
| % $t = s$[7] | 0% | 0% | 0% | 0% |
| $w$ | 2 | 1.75 | 2.5 | 2.25 |
| Target EMA rate | 0.99 | 0.99 | 0.99 | 0.99 |
| Eval EMA rate[8] | 0.9999 | 0.9999 | 0.9999 | 0.9999 |
| Label dropout | 0.1 | 0.1 | 0.1 | 0.1 |

Table 6: Experimental settings for different architectures and datasets.

Therefore,

$$\left\| \frac{d}{dt} \mathbf{f}_\theta(\mathbf{x}_t, t, s) + \mathbf{u}(\mathbf{x}_t, t) \right\|_2^2 \tag{91}$$

$$= \left\| - \mathbf{F}_\theta(\mathbf{x}_t, t, s) + \underbrace{(s-t)\Big[ \mathbf{u}(\mathbf{x}_t, t) \cdot \nabla_{\mathbf{x}_t} \mathbf{F}_\theta(\mathbf{x}_t, t, s) + \partial_t \mathbf{F}_\theta(\mathbf{x}_t, t, s) \Big] + \mathbf{u}(\mathbf{x}_t, t)}_{F_{\text{tgt}}} \right\|^2 \tag{92}$$

which is the MeanFlow loss.

### E.2 FLOW MAP SELF-DISTILLATION

Boffi et al. (2025) proposes three different self-distillation approaches for training flow maps from scratch.

## F ADDITIONAL EXPERIMENT DETAILS

We present the overall training details in Table 6

---

[5]Except for zero-init layers in AdaLN-Zero.

[6]All time embedding MLP layers before input into DiT blocks.

[7]This means the percentage of setting $t = s$, *e.g.* 0% means both loss terms exist at all times.

[8]EMA used for final evaluation, separate from the EMA used for training target.

### F.1 Architecture and Optimization

**VAE.** We follow Zhou et al. (2025) for the VAE setting, which uses the standard Stable Diffusion VAE (Rombach et al., 2022) but with a different scale and shift. Please refer to the paper for details.

**Architecture.** All architecture decisions follow DiT (Peebles & Xie, 2023) except for the changes described in the main text. For our XL-sized model, we follow DiT-XL and use 1152 hidden size but use 18 heads instead of 16 heads. This is purely for efficiency reasons because 18 heads under 1152 total hidden size implies head dimension is 64, while the original 16 heads result in head dimension 72. Flash attention JVP's runtime is sensitive to redundancy in memory allocations. As 64 is a power of 2 our kernel can fully allocate appropriately sized CUDA blocks, while 72 leaves significant chunks unused. We observe that the original 16-head decision is $\times 1.25$ slower than the 18-head variant. In comparing FID of the two versions, we observe they perform similarly throughout training.

Following Zhou et al. (2025), we use $t - s$ as our second time condition into the architecture rather than directly injecting $s$. For injecting $w$, we follow Chen et al. (2025) and use $\beta = 1/w$ as our condition, and if random CFG is used training, we sample $\beta \sim \mathcal{U}(\frac{1}{w_{\max}}, \frac{1}{w_{\min}})$ and set $w = 1/\beta$. Note that Chen et al. (2025) uses $\beta \sim \mathcal{U}(0, 1)$ which amounts to $w_{\min} = 1$ and $w_{\max} = \infty$, but arbitrarily large $w$ is never used in practice so $w_{\max}$ can be set to a realistic finite value.

**Optimization.** Besides setting $\beta_2 = 0.95$, we follow the default optimizer used by DiT and optimize with BF16 precision. We de not use any learning rate scheduler.

### F.2 Details on Random CFG with MeanFlow

In MeanFlow (Geng et al., 2025), the authors introduce a mixing scale $\kappa$ such that the field with guidance scale $w$ is given by

$$\mathbf{v}(\mathbf{x}_t, t, c, w) = w\mathbf{v}_t + \kappa\mathbf{u}_\theta(\mathbf{x}_t, t, c, w) + (1 - w - \kappa)\mathbf{u}_\theta(\mathbf{x}_t, t, w) \tag{93}$$

It specifies that the effective guidance scale is $w' = \frac{w}{(1-\kappa)}$. This is because since $\mathbf{u}_\theta(\mathbf{x}_t, t, c, w) \approx \mathbf{v}(\mathbf{x}_t, t, c, w)$, rearranging it to LHS and dividing both sides by $(1 - \kappa)$ gives

$$(1 - \kappa)\mathbf{v}(\mathbf{x}_t, t, c, w) = w\mathbf{v}_t + (1 - w - \kappa)\mathbf{u}_\theta(\mathbf{x}_t, t, w) \tag{94}$$

$$\mathbf{v}(\mathbf{x}_t, t, c, w) = \frac{w}{(1 - \kappa)}\mathbf{v}_t + (1 - \frac{w}{(1 - \kappa)})\mathbf{u}_\theta(\mathbf{x}_t, t, w) \tag{95}$$

This constrains $\kappa \in [0, 1)$. However, in the case of random CFG, to make use of $\mathbf{u}_\theta(\mathbf{x}_t, t, c, w)$, we try the simple linear mixing (the default CFG reweighting)

$$\mathbf{v}(\mathbf{x}_t, t, c, w + \kappa) = w\mathbf{v}_t + \kappa\mathbf{u}_\theta(\mathbf{x}_t, t, c, 1) + (1 - w - \kappa)\mathbf{u}_\theta(\mathbf{x}_t, t, 1) \tag{96}$$

where $w$ and $\kappa$ are both randomly sampled with finite boundaries. In this case $\mathbf{u}_\theta(\mathbf{x}_t, t, c, 1) \not\approx \mathbf{v}(\mathbf{x}_t, t, c, w + \kappa)$ and thus $\kappa$ is not constrained to be smaller than 1. When $w = 0$, it becomes regular CFG with network approximation of the CFG velocity, and when $\kappa = 0$ it becomes MeanFlow CFG with $\mathbf{v}_t$ approximation of the CFG velocity. This construction subsumes both implementation cases. In our experiments, we use $\kappa \sim \mathcal{U}(0, c_{\max}), w \sim \mathcal{U}(1, c_{\max})$ for some constant $c_{\max}$. However, we acknowledge that this observed training fluctuation may depend on exact training settings and environments, and may be fixable via empirical tricks such as adjusting AdamW parameters or gradient clipping, etc. We present the training in the simplest settings without such tricks to best illustrate our point.

### F.3 CFG-Conditioned Flow Matching

As in our method, we similarly observe tradeoff in FID if FM is trained to condition on CFG scale $w$ with randomly sampled $w$ during training (Chen et al., 2025). During inference time, $w$ is injected into the network so that the CFG velocity field can be approximated by a single forward call. We inject $w$ using positional embedding just like the diffusion time, and during training we sample $\beta \sim \mathcal{U}(0, 1)$ and

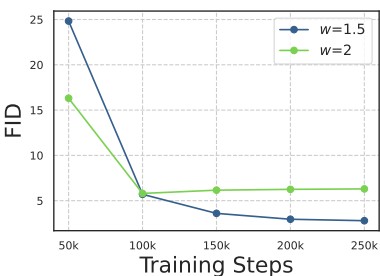

Figure 12: $w$-conditioned FM training experiences tradeoff.

set $w = 1/\beta$, following Chen et al. (2025). We show in Figure 12 that as the model trains, the FID of $w = 1.5$ decreases but $w = 2$ increases for later training steps. This tradeoff is similarly observed in our method as presented in the main text.

## F.4 ADDITIONAL VISUAL SAMPLES

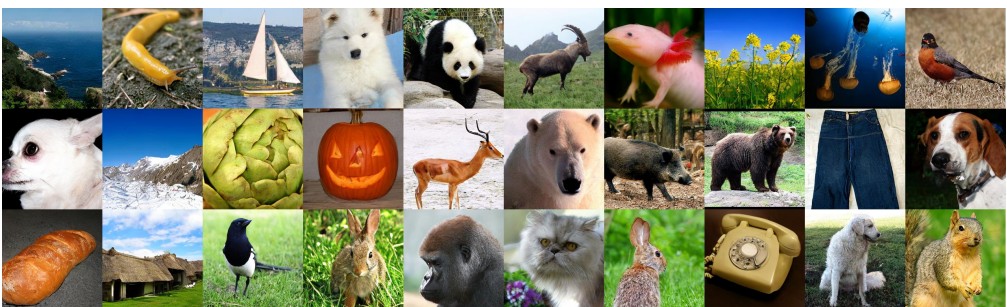

Figure 13: Additional ImageNet-256×256 samples from 1-NFE TVM model.

