# OpenReview forum: "Terminal Velocity Matching"
_ICLR.cc/2026/Conference — ICLR 2026 Poster_

### Official Review · Reviewer_EXtX · 2025-10-27

**Soundness:** 3
**Presentation:** 3
**Contribution:** 3
**Rating:** 6
**Confidence:** 4

**Summary:**

The paper proposes a novel method for training one or few-step generative models by matching the terminal velocity. Through a clever rearrangement of the displacement equation between arbitrary points t and s along the ODE trajectory, the paper derives a novel objective which contains the instantaneous velocity at s and the generic Flow Matching loss. The formulation requires high order gradients through the JVP operation, but proposes improvements to the JVP computation to ensure scalability. The method further introduces improvements to DiT architecture and a way to include CFG conditioning. The method is tested on the common benchmark Imagenet 256x256 and achieves SOTA results for one and few-step generation.

**Strengths:**

The loss formulation is novel and well motivated, the paper introduces several relevant contribution such as DiT modification, scalable JVP computation, novel CFG training procedure. The results are strong and the method is scalable.

**Weaknesses:**

To me a weakness of the method it is hard to tell from the manuscript to what we can attribute the good performance. The overall results are great so there is no doubt that the proposed method is effective, but I wonder for example what results one could get from MeanFlow when using the improved DiT architecture. TVM is good but introduces some complications, such as using a EMA version of the model in the target, and having to backpropagate through JVP. While the latter is addressed, I wonder if just the improved architecture can work well on MeanFlow. I checked through the text and it doesn't specify whether meanflow is trained with standard DiT, so I am not sure what the retrained results from table 1 correspond to.

A common benchmark for these models is CIFAR10 but that was not reported. While it is a relatively small dataset, it can provide useful insight on the performance of the model without CFG, as well as on the sensitivity of the model to specific hyperparameters.

Overall, I think this method is promising, but seems like many details are left out, which makes it hard to fully understand the significance of the contribution (see questions). Also there is no open source code.

**Questions:**

- 1) Maybe I missed it but I could not find the exact number of training iterations for the results in table 3, same for batch size.
- 2) There also seem to be no info about the total training time. The method uses a forward pass with JVP plus a standard forward pass with the EMA model, and backward pass through JVP. Intuitively this should be much slower than MeanFlow or sCT, and it would be interesting to understand by how much.

---

> ### Author Response · Authors · 2025-11-20
> **Rebuttal 1/2**
>
> We thank the reviewer for the generally positive feedback and insightful comments. We wish to address your concerns below.
>
> > what we can attribute the good performance. The overall results are great so there is no doubt that the proposed method is effective, but I wonder for example what results one could get from MeanFlow when using the improved DiT architecture. TVM is good but introduces some complications, such as using a EMA version of the model in the target, and having to backpropagate through JVP. While the latter is addressed, I wonder if just the improved architecture can work well on MeanFlow. I checked through the text and it doesn't specify whether meanflow is trained with standard DiT, so I am not sure what the retrained results from table 1 correspond to.
>
>  We first want to stress that the improved designs come from the unique insight provided by our theory and are a unique contribution of our work, even though they are simple enough to be generally applicable. However, we understand the concern and conducted a preliminary MeanFlow study with our improvements. We did not have enough time to train till completion for the time being and report our findings at 150K iteration
>
> |     | 1-NFE     | 2-NFE   |
> |---|---|---|
> | w/ naive DiT-XL/2        |    8.94     |   6.73   |
> | w/ improved DiT-XL/2  |     **8.89**      | **5.88**  |
>
> With our proposed changes 1 NFE performs similarly and 2 NFE performs better, so the semi-Lipschitz control indeed similarly helps with MeanFlow. However, again, we believe this is a unique contribution of our theory and for this reason we keep the baselines as originally proposed in the literature and treat our method with semi-Lipschitz control as a single new approach.
>
>  *Update: our MeanFlow training run above has saturated its 1-NFE performance at 3.66 FID with improved DiT. This is marginally worse than the reported number and our previous retrained run 3.43 FID. Therefore, we conclude that the TVM performance cannot be entirely explained by architecture changes.*
>
>
> > A common benchmark for these models is CIFAR10 but that was not reported. While it is a relatively small dataset, it can provide useful insight on the performance of the model without CFG, as well as on the sensitivity of the model to specific hyperparameters.
>
> In the limited time frame we quickly tested TVM on CIFAR10 with the default DDPM++ architecture and achieved 3.50 one-step FID. We follow sCT for time sampling distribution and general training settings. Please note that we had limited time to tune on this dataset among all other experiments to run, so this result is very suboptimal. However, this shows great potential for TVM to be generally applicable. We note that although UNet architecture may suffer from the same Lipschitzness issue due to its use of SDPA and GroupNorm, we find that in practice they do not cause significant instability for CIFAR10. We hypothesize that this is due to relatively small model and data size.
>
> Regarding concerns on sensitivity to hyperparameters, we have performed extensive ablation on all design choices in our Experiment section. From our experience, we do not find TVM is sensitive to any specific parameters as long as Lipschitz-control is performed as suggested. However, we do identify some important choices that enhance convergence and quality: (1) EMA target and (2) gap time sampling. For thesis choices, as long as one uses EMA with rate >=0.9 and <=0.999, and gap time sampling with logit-normal distribution as presented in Appendix F, small variations in these hyperparams generally give similar results.
>
>
>
> > Overall, I think this method is promising, but seems like many details are left out, which makes it hard to fully understand the significance of the contribution (see questions). Also there is no open source code.
>
> We have updated our submission to include all training details. Please see Appendix F. And we will release code upon acceptance.
>
> > Maybe I missed it but I could not find the exact number of training iterations for the results in table 3, same for batch size.
>
> Batch size for all the ablation studies is noted in Ablation Studies section. They trained with 200K steps with batch size 1024.

---

> > ### Author Response · Authors · 2025-11-20
> > **Rebuttal 2/2**
> >
> > > There also seem to be no info about the total training time. The method uses a forward pass with JVP plus a standard forward pass with the EMA model, and backward pass through JVP. Intuitively this should be much slower than MeanFlow or sCT, and it would be interesting to understand by how much.
> >
> > Thank you for noting the computation cost. We conducted study on per-train-step runtime and per-GPU memory cost below. This is similarly updated in our revision.
> >
> > Since PyTorch SDPA package does not officially support JVP, the following default MeanFlow setting uses naive SDPA, which is the only way currently to perform JVP by default in PyTorch. Both statistics are averaged over 10 training steps, and EMA updates are excluded from the calculations.
> >
> > **per-GPU Peak Memory Usage**
> >
> > | Batch Size   |  MeanFlow (w/ naive SDPA)   | MeanFlow (w/ our kernel) |      TVM (w/ improved DiT)      | TVM (w/ naive DiT )  |  TVM (w/ improved DiT, detach JVP) |
> > |---|---|---|---|---|---|
> > | 128 |  50.30 GB   |   **31.31 GB**   |   43.21 GB   |   38.85 GB  |   36.17 GB  |
> > | 256 |  OOM         |    **46.73 GB**   |   71.44 GB  |  59.53 GB |   55.71 GB      |
> >
> >
> > **Averaged per-Step Runtime**
> >
> > | Batch Size   |  MeanFlow (w/ naive SDPA)   | MeanFlow (w/ our kernel)  |     TVM (w/ improved DiT)       |  TVM (w/ naive DiT) | TVM (w/ improved DiT, detach JVP) |
> > |---|---|---|---|---|---|
> > | 128 |  0.76 s   |  0.77 s     |  0.69 s   |   0.60 s   |   **0.51 s**     |
> > | 256 |  -     |   0.81 s    |    0.95 s  |  0.86 s   |    **0.69**   |
> >
> > First we note that with naive SDPA MeanFlow runs out of memory on batch size 256, and on 128 batch size it takes 50.30GB which is extremely inefficient. We next ablate by substituting the attention with our kernel and the memory usage is only 60% of the baseline without sacrificing runtime. Our kernel gives significant boost in efficiency. In comparison to improved MeanFlow, TVM utilizes more memory due to JVP backward pass, but it still outperforms naive MeanFlow in memory and runtime. With batch size 256, TVM with improved DiT has slower runtime and uses more memory compared to optimized MeanFlow. We find that this is actually mostly due to more architectural changes proposed, because as we test TVM with naive DiT, we notice both memory and runtime are marginally higher than improved MeanFlow. Lastly, if runtime is a concern, we can also simply detach JVP as in MeanFlow which biases its gradient but gives significant speedup, surpassing MeanFlow step time.
> >
> > Despite slower per-step runtime (with full TVM setup) compared to MeanFlow (with our kernel), TVM converges at ~4700 GPU-hours while MeanFlow converges at ~4400 GPU-hours. The total runtime is about 5% more, which is marginal in practice.
> >
> >
> >
> >
> > Please let us know if you have any more concerns and we are happy to address them.

---

> ### Comment · Reviewer_EXtX · 2025-11-26
>
> I thank the authors for addressing my concerns. After reviewing the responses to all reviewers, I have decided to keep my score unchanged. While I am convinced of the significance of the contributions, I believe the paper would benefit from a more comprehensive discussion and evaluation, some of which has been partially clarified in the rebuttal.
> I understand that the limited timeframe makes it challenging to run extensive and compute-intensive experiments, and I appreciate the authors’ effort to provide as many results as possible within these constraints.

---

### Official Review · Reviewer_ep7Z · 2025-10-30

**Soundness:** 3
**Presentation:** 3
**Contribution:** 3
**Rating:** 6
**Confidence:** 5

**Summary:**

The authors propose Terminal Velocity Matching (TVM), a few-step generation framework that matches terminal-time velocities instead of initial-time ones. To make TVM scalable for transformer architectures, the paper introduces minimal architectural modifications (e.g., RMSNorm-based AdaLN, Lipschitz initialization) and an efficient Flash-Attention JVP kernel that supports backward passes through Jacobian–Vector Products (JVPs). Combined with a simple training recipe—without curriculum training schedules or adaptive weighting—TVM achieves strong results on ImageNet-256×256, setting a new Pareto frontier for few-step generative models (FID = 3.30 @ 1-NFE, 2.49 @ 2-NFE) while maintaining training stability.

**Strengths:**

1. **Boundary condition at terminal time**

By enforcing velocity matching at the terminal rather than initial timestep, the method avoids evaluating JVPs involving guided velocities that often exhibit large norms and high variance during training. This could be more beneficial for stabilizing training when scaling to larger dimensionality, where the guided velocities often exhibit even larger norms and higher variance.

2. **Simple and stable training recipe**

The paper achieves stable one-stage training with only minor modifications to model architecture and without requiring any curriculum schedule, auxiliary objectives, or adaptive weighting heuristics. These simplifications make the approach both reproducible and again appealing for scaling to large models or high-dimensional data.

3. **Customized JVP kernel supporting efficient backpropagation**

The authors develop a fused Flash-Attention JVP kernel that supports backward passes on Jacobian–Vector Products within transformer blocks. This kernel significantly reduces memory usage and improves runtime efficiency during training. This engineering contribution is non-trivial and essential for making TVM practical at scale.

**Weaknesses:**

1. **Backpropagation through JVP.**

Unlike prior continuous-time consistency models where JVP terms are detached from the gradient graph (sCM, MeanFlow, etc.), TVM explicitly backpropagates through the JVP term introducing additional computational cost. This could become prohibitive for large-scale models. Providing quantitative analysis (e.g., runtime, memory, or gradient-cost overhead relative to MeanFlow) would help stress the concern.

2. **Insufficient ablations.**

While the design choices are well-motivated, the paper would benefit from deeper empirical validation. For instance, ablations comparing training with vs. without adaptive weighting on the training loss, the convergence of the original DiT versus the Lipschitz-controlled variant, and MeanFlow vs. TVM under identical random classifier-free guidance (CFG) conditions. Reporting gradient-norm / JVP-norm comparisons between TVM and MeanFlow would further support the claim of reduced JVP instability. Including recent baselines such as FACM (Peng et al., 2025) would also improve completeness.

3. **Need for deeper insight behind ablation trends,**

The empirical results raise several interesting but unexplained phenomena. For example, using any proportion of $t=s$ (pure flow-matching loss) degrades 1-step performance, and employing EMA targets accelerates convergence. A deeper discussion of why these trends arise would enrich the paper’s interpretability and practical guidance.

Peng, Yansong, Kai Zhu, Yu Liu, Pingyu Wu, Hebei Li, Xiaoyan Sun, and Feng Wu. "Flow-anchored consistency models." arXiv preprint arXiv:2507.03738 (2025).

**Questions:**

1. **Distributional guarantees vs. flow-map loss.**

The authors mention that Flow Map methods lack explicit distributional control, whereas TVM provides a 2-Wasserstein control. Could the authors elaborate on the difference? At a glance, the TVM loss resembles the Lagrangian distillation loss proposed in Flow Map—what key term enforces distributional regularity here?

2. **Effect of removing forced FM loss.**

TVM achieves superior 1-step generation performance even with 0% enforced Flow Matching samples. What intuition do the authors have for why completely removing the auxiliary FM supervision leads to better convergence?

3. **EMA acceleration.**

What is the authors’ intuition behind the observed acceleration of convergence when using EMA weights for the training objective? Does this act primarily as a variance-reduction mechanism?

4. **Distillation potential.**

Have the authors explored or considered using TVM as a distillation objective? Since TVM naturally supports terminal-condition regularization, it could serve as a stable distillation formulation for distilling multi-step teacher models into few-step generators.

---

> ### Author Response · Authors · 2025-11-20
> **Rebuttal 1/3**
>
> We thank the reviewer for generally positive feedback. And we appreciate your further suggestions and questions. We wish to clarify below.
>
> > Unlike prior continuous-time consistency models where JVP terms are detached from the gradient graph (sCM, MeanFlow, etc.), TVM explicitly backpropagates through the JVP term introducing additional computational cost. This could become prohibitive for large-scale models. Providing quantitative analysis (e.g., runtime, memory, or gradient-cost overhead relative to MeanFlow) would help stress the concern.
>
> Thank you for expressing your concern about efficiency. To address this, we conduct the following ImageNet-256x256 experiments to compare the peak memory usage and per-step runtime on a single 8-GPU NVIDIA H100 node with PyTorch.
>
> Since PyTorch SDPA package does not officially support JVP, the following default MeanFlow setting uses naive SDPA, which is the only way currently to perform JVP by default in PyTorch. Both statistics are averaged over 10 training steps, and EMA updates are excluded from the calculations.
>
> **per-GPU Peak Memory Usage**
>
> | Batch Size   |  MeanFlow (w/ naive SDPA)   | MeanFlow (w/ our kernel) |     TVM (w/ improved DiT)      | TVM (w/ naive DiT )  |  TVM (w/ improved DiT, detach JVP) |
> |---|---|---|---|---| ---|
> | 128 |  50.30 GB   |   **31.31 GB**   |   43.21 GB   |   38.85 GB  |   36.17 GB  |
> | 256 |  OOM         |    **46.73 GB**   |   71.44 GB  |  59.53 GB |   55.71 GB      |
>
>
> **Averaged per-Step Runtime**
>
> | Batch Size   |  MeanFlow (w/ naive SDPA)   | MeanFlow (w/ our kernel)  |     TVM  (w/ improved DiT)        |  TVM (w/ naive DiT) | TVM (w/ improved DiT, detach JVP) |
> |---|---|---|---|---| ---|
> | 128 |  0.76 s   |  0.77 s     |  0.69 s   |   0.60 s   |   **0.51 s**     |
> | 256 |  -     |   0.81 s    |    0.95 s  |  0.86 s   |    **0.69**   |
>
>
> First we note that with naive SDPA MeanFlow runs out of memory on batch size 256, and on 128 batch size it takes 50.30GB which is extremely inefficient. We next ablate by substituting the attention with our kernel and the memory usage is only 60% of the baseline without sacrificing runtime. Our kernel gives significant boost in efficiency. In comparison to improved MeanFlow, TVM utilizes more memory due to JVP backward pass, but it still outperforms naive MeanFlow in memory and runtime. With batch size 256, TVM with improved DiT has slower runtime and uses more memory compared to optimized MeanFlow. We find that this is actually mostly due to more architectural changes proposed, because as we test TVM with naive DiT, we notice both memory and runtime are marginally higher than improved MeanFlow. Lastly, if runtime is a concern, we can also simply detach JVP as in MeanFlow which biases its gradient but gives significant speedup, surpassing MeanFlow step time.
>
> Despite slower per-step runtime (with full TVM setup) compared to MeanFlow (with our kernel), TVM converges at ~4700 GPU-hours while MeanFlow converges at ~4400 GPU-hours. The total runtime is about 5% more, which is marginal in practice.

---

> > ### Author Response · Authors · 2025-11-20
> > **Rebuttal 2/3**
> >
> > > While the design choices are well-motivated, the paper would benefit from deeper empirical validation. For instance, ablations comparing training with vs. without adaptive weighting on the training loss, the convergence of the original DiT versus the Lipchitz-controlled variant, and MeanFlow vs. TVM under identical random classifier-free guidance (CFG) conditions. Reporting gradient-norm / JVP-norm comparisons between TVM and MeanFlow would further support the claim of reduced JVP instability. Including recent baselines such as FACM (Peng et al., 2025) would also improve completeness.
> >
> > We address these components one by one.
> >
> > **Adaptive-weighting**:
> >
> > We assume that the reviewer refers to the adaptive weighting introduced in MeanFlow. We believe that this trick does not fit well with our framework as adaptive weighting will deviate from our 2-Wasserstein interpretation, which utilizes only MSE loss. However, we still conduct experiments using adaptive weighting on our full TVM objective with DiT-B architecture for simplicity. We present training progress results evaluated on 1-step FID below
> >
> > |     |  w/o adaptive weighting  |  w/ adaptive weighting  |
> > |---|---|---|
> > |  50k iters |   **43.15** |    46.27    |
> > |  100k iters |   **23.34**  |  25.20      |
> >
> > We find that adaptive weighting is in fact slightly worse for TVM, so we leave it out of our design space for simplicity.
> >
> > **Convergence of original DiT**:
> >
> > As noted in Section 4, Figure 4, naive DiT will experience significant instability due to the attention module’s lack of Lipchitz continuity. While we find that using AdamW $\beta_2=0.95$ can bypass the instability at ImageNet’s scale, as is similarly adopted by MeanFlow, we believe this does not solve the fundamental issue and may prevent further scaling. Nevertheless, we conducted a comparison with a naive DiT under $\beta_2=0.95$.
> >
> > |     |  naive DiT  |   Lipschitz-controlled DiT   |
> > |---|---|---|
> > |  100k iters |  5.07  |   **4.20**      |
> > |  200k iters |  3.41   |   **3.35**       |
> >
> > We find that the two versions work similarly well as training goes on and Lipschitz-controlled DiT is marginally better probably due to improved architecture smoothness. Without $\beta_2=0.95$, we stress that naive DiT will completely fail while our version remains stable. We believe the need for Lipschitz-control is a crucial insight as illustrated by our theory.
> >
> >
> > **MeanFlow/TVM under random CFG condition**:
> >
> > We have updated the corresponding figure in our revision to include both MeanFlow grad norm and TVM grad norm as well as comparisons of $u(x_t,t)$ norm. It is clearly shown that TVM grad norm is much smoother and lower than MeanFlow, and TVM $u(x_t,t)$ norm as given by the proxy network has much lower variance than MeanFlow under the same random CFG settings. Note that MeanFlow and TVM have very different $u(x_t,t)$ norm in this case, and this is because MeanFlow training has collapsed under random CFG settings.
> >
> > **Other baselines such as FACM**:
> >
> > We appreciate the reference. This is a very interesting work and achieves impressive results on ImageNet. However, it seems that this is a concurrent work with ours and is also submitted to ICLR 2026 so we refrain from directly comparing it as a baseline for our work. However, we will include its discussion in our related works.
> >
> >
> >
> >
> > > The authors mention that Flow Map methods lack explicit distributional control, whereas TVM provides a 2-Wasserstein control. Could the authors elaborate on the difference? At a glance, the TVM loss resembles the Lagrangian distillation loss proposed in Flow Map—what key term enforces distributional regularity here?
> >
> > Thank you for bringing up this point!  First of all, Lagrangian distillation loss in Flow Map Matching is a distillation objective and assumes that the ground-truth velocity field is exactly approximated by a teacher model. However, it does not account for the error between the ground-truth velocity and the teacher velocity. Its original bound in the distillation setting only bounds distributional divergence between *teacher* and student, rather than between student and the data. This is crucial in the training-from-scratch setting because in early training the pseudo-teacher (i.e. parameterized by the student itself) does not well-approximate the ground-truth velocity. And this additional discrepancy is accounted for in our second loss term (i.e. the FM loss). It can be shown that the simple addition of these two loss terms with proper weightings explicitly bounds 2-Wasserstein distance between the student and *data*, assuming that our model is Lipchitz-continuous.

---

> ### Author Response · Authors · 2025-11-20
> **Rebuttal 3/3**
>
> > TVM achieves superior 1-step generation performance even with 0% enforced Flow Matching samples. What intuition do the authors have for why completely removing the auxiliary FM supervision leads to better convergence?
>
> We think there may be a misunderstanding on this ablation. The 0% here means 0% chance of setting $t=s$ which means both loss terms exist at all times. FM loss will always exist; the percentage only controls how often the first term is used. This finding suggests that additionally setting $t=s$ for some probability (as is necessarily the case for MeanFlow) is an unnecessary design choice for us since the vanilla loss terms together already achieves superior results.
>
> > What is the authors’ intuition behind the observed acceleration of convergence when using EMA weights for the training objective? Does this act primarily as a variance-reduction mechanism?
>
> This is correct. The motivation is two-fold – for both quality boost and variance reduction. First, it is widely accepted that EMA weights in FM produce much higher-quality samples than non-EMA weights. Using these as regression target leads to higher-quality results. Second, EMA updates slower and is less prone to optimization noise, thus facilitating convergence.
>
>
> > Have the authors explored or considered using TVM as a distillation objective? Since TVM naturally supports terminal-condition regularization, it could serve as a stable distillation formulation for distilling multi-step teacher models into few-step generators.
>
>
> TVM can indeed be used for distillation purposes. More specifically, the full objective as is can serve as a *finetuning* objective on any trained FM model. The difference from a distillation objective is that the FM model to be initialized from does not need to fully converge before being used because full TVM objective guarantees convergence towards data distribution. To be used as a pure distillation objective, simply remove the FM term and use a pretrained teacher as the regression target.
>
>
> Please let us know if you have any more concerns and we are happy to address them.

---

### Official Review · Reviewer_4c17 · 2025-10-31

**Soundness:** 3
**Presentation:** 3
**Contribution:** 2
**Rating:** 4
**Confidence:** 4

**Summary:**

This paper introduces Terminal Velocity Matching (TVM), a new training objective for generative modeling that generalizes Flow Matching (FM) by matching velocity fields at terminal rather than initial points along probability flow trajectories. The authors prove that the TVM loss upper bounds the 2-Wasserstein distance between data and model distributions, providing a principled connection between training dynamics and optimal transport theory. They also propose practical architectural modifications—such as semi-Lipschitz normalization, FlashAttention-based Jacobian-vector products, and scaled parameterization—that stabilize training and enable few-step sampling. Empirically, TVM achieves state-of-the-art single- and few-step image generation performance on ImageNet-256 with a DiT backbone.

**Strengths:**

1. The theoretical formulation is elegant and well-motivated, linking Flow Matching to a Wasserstein upper bound through terminal velocity constraints.
2. The method achieves excellent efficiency–quality trade-offs, outperforming existing one-step and few-step baselines (e.g., Consistency Models, MeanFlow) on ImageNet-256.
3. The architectural refinements (semi-Lipschitz normalization, FlashAttention JVP) are practically valuable contributions that could generalize to other diffusion or flow-based frameworks.

**Weaknesses:**

1. While theoretically grounded, the intuition behind “terminal velocity” could be elaborated further—especially how it differs in practice from midpoint or integral matching.
2. The scope of evaluation is limited to class-conditional ImageNet-256. Demonstrating robustness on higher-resolution or unconditional datasets (e.g., ImageNet-512, COCO) would strengthen generality claims.
3. The paper relies on a single architecture (DiT-XL/2). It is unclear whether TVM’s benefits extend to U-Net–based or continuous-time flow models.
4. The FlashAttention JVP optimization, though valuable, feels somewhat tangential to the conceptual contribution and could be relegated to an appendix.
5. Despite the theoretical connection to Wasserstein distance, there is no empirical verification of this bound (e.g., via transport cost evaluation), leaving its practical tightness untested.

**Questions:**

1. Can the authors clarify why terminal velocity matching improves convergence over initial or midpoint matching?
2. How tight is the Wasserstein upper bound empirically—has this been tested or approximated?
3. Is the FlashAttention JVP essential for convergence, or mainly for speed?
4. Could the method generalize to other architectures or modalities (e.g., U-Nets, text-to-image, or video)?

---

> ### Author Response · Authors · 2025-11-20
> **Rebuttal 1/3**
>
> We thank the reviewer for all the insightful questions. We wish to address them one by one below.
>
> > While theoretically grounded, the intuition behind “terminal velocity” could be elaborated further—especially how it differs in practice from midpoint or integral matching.
>
> The intuition can be understood as follows. Imagine an ODE path in space and one wants the model to directly map from any intermediate time $t$ to time $0$ (the time where data lies) on this ODE path. The one-step model path is a straight path connecting a starting point $x_t$ to an endpoint $x_s$ where $x_s$ is the ODE solution from $x_t$. The model path connecting $x_t$ to $x_0$ will be equivalent to the ground-truth ODE from $x_t$ to $x_0$ if the *velocity* of the model at $s$ is the same as the velocity of the ground-truth ODE at $s$.  If this equality holds true for all $s\in [0,t]$, one can imagine a velocity vector (i.e. the true marginal velocity) guiding the terminal point of the model path in space from $t$ to $0$ tracing the exact ODE path and landing on $x_0$.
>
> This is crucially different from directly matching integral because direct integral matching will require explicit ODE simulation during training. We can bypass integral matching by matching their terminal velocity instead. In fact, we show our terminal velocity error upper bounds direct integral matching error (see Eq. 7).
>
> For midpoint matching, if we understand correctly, the reviewer refers to using midpoint rule as an approximation for the integral. This is less ideal because although this bypasses explicit integration, it still requires N evaluations of the velocity model for N midpoints and the approximation error becomes large with coarser discretization. Matching velocity, however, guarantees the net displacement will match due to the fundamental theorem of calculus.
>
> > The scope of evaluation is limited to class-conditional ImageNet-256. Demonstrating robustness on higher-resolution or unconditional datasets (e.g., ImageNet-512, COCO) would strengthen generality claims.
>
> Please see updated revision for results on ImageNet-512x512. In short, we outperform other training-from-scratch few-step methods such as sCT and MeanFlow and achieve 4.32 1-NFE FID. TVM can also surpasse baseline diffusion with 4 NFEs, achieving 2.94 FID vs. 3.04 FID (500-NFE DiT-XL/2). Please also note that we updated performance on ImageNet-256x256 where we can achieve 3.29 1-NFE FID and 1.99 4-NFE FID, matching and exceeding diffusion baselines (2.27 FID). Please also see supplementary materials for preliminary TVM results on Text-to-Image trained at 10B+ scale. We leave out the training details as it is beyond our scope of the current paper.
>
> > The paper relies on a single architecture (DiT-XL/2). It is unclear whether TVM’s benefits extend to U-Net–based or continuous-time flow models.
>
> We want to stress that our main theory sheds most light on the flaws of current diffusion transformers as inspired by prior literature on transformers themselves (i.e. their lack of Lipchitz constraints). And since most large-scale models default to DiT-style architecture we believe it is most appropriate to focus on DiT as the backbone. Nevertheless, similar arguments apply to UNet architectures. Most popular UNets (e.g. EDM UNet) also suffer from lack of Lipchitz continuity due to its use of dot-product self-attention and its use of group normalization (similar to LayerNorm). However, empirically, we notice that naive DDPM++ works without instability and in the limited time frame given we were able to achieve 3.50 1-NFE FID for CIFAR10. We attribute the lack of instability to relatively small model size and data size, and given the limited time to tune on CIFAR10, it shows TVM’s great potential to be generally applicable to different architecture and domains.
>
>
>
> > The FlashAttention JVP optimization, though valuable, feels somewhat tangential to the conceptual contribution and could be relegated to an appendix.
>
> We argue that our theoretical contribution is tightly coupled with our empirical contribution because to train with the proposed objective at scale it is unavoidable to solve backpropagation through JVP (as proposed by the objective). Otherwise we will deviate from our proposed objective. Additionally, backpropagating through JVP is novel to the community so we believe it is best to highlight the new kernel as one of our main contributions, especially when it enables training at scale.

---

> > ### Author Response · Authors · 2025-11-20
> > **Rebuttal 2/3**
> >
> > > Despite the theoretical connection to Wasserstein distance, there is no empirical verification of this bound (e.g., via transport cost evaluation), leaving its practical tightness untested.
> >
> > We want to emphasize that our goal for the theory is not to propose an exact computable bound on Wasserstein distance but to use the bound to justify the theoretical soundness of our objective in terms of distribution matching and to additionally serve as a practical guideline. Specifically, (1) most one/few-step methods trained from scratch such as MeanFlow, sCT, etc. lack distributional guarantee, and our theory is the first to show our training objective explicitly upper bounds distribution divergence with proper weightings. (2) The theory serves as an empirical guide for designing better architectures as it sheds light on critical flaws of the most popular diffusion transformers, i.e. the lack of Lipchitz smoothness.
> >
> > Since the bound depends on the Lipchitzness of the architecture which is difficult to compute in general, we leave the empirical computation to future work. Note, however, many optimization techniques (e.g. Muon) and architectural designs aim to constrain the Lipchitzness of architecture, so the bound becomes tighter with better optimization and architectures, but these are beyond the scope of the current work.
> >
> >
> > > Can the authors clarify why terminal velocity matching improves convergence over initial or midpoint matching?
> >
> > We are uncertain what exactly is meant by initial and midpoint matching by the reviewer in this context, but we make our best effort to interpret what is meant. Please correct us if we are wrong.
> >
> > We assume that by initial point matching the reviewer means MeanFlow/sCT-style learning the velocity at the initial time $t$. For these methods, as shown in Eq. 18, requires marginal velocity $u(x_t, t)$ which in practice is substituted with the conditional velocity $v_t$. Although it can be shown that this substitution, along with stop-grad operation, results in the same expected gradient as the case when $u(x_t, t)$ is known, $v_t$ introduces additional variance during optimization because $v_t$ is also propagated through JVP for obtaining the training target.
> >
> > In contrast, TVM uses our own network output as the training target and does not experience additional variance given any $x_s$. No conditional velocity is used to train JVP and the optimization process is less noisy. With EMA network as the learning target, optimization variance is additionally reduced due to EMA’s slow update nature.
> >
> > Similarly, for midpoint matching we assume the reviewer means using a middle velocity in the integral path as proxy for the integral results. The convergence of such methods will depend on how the midpoint is chosen. One can randomly sample this midpoint or deterministically choose a midpoint. If random sampling is used, additional variance is introduced for optimization, and if deterministic midpoints are used, the approximation is highly biased which may hurt performance. TVM (specifically the first loss term) is both accurate and variance-free because it exactly upper bounds path-error (Eq. 7) and its training target has no randomness given any initial $x_t$.
> >
> >
> > > How tight is the Wasserstein upper bound empirically—has this been tested or approximated?
> >
> >
> > As we mentioned previously, tightness of the bound is not a primary focus of this work as our theory only serves to (1) provide connection to distributional guarantee in contrast to many previous one/few-step methods from scratch (e.g. MeanFlow, sCT, etc.) and (2) reveals flaws about current architectures and serves as guidelines for designing better architectures with Lipchitz controls (which inspires the simple architectural changes in this paper). However, it would be interesting for future work to investigate the tightness of the bound in relation to the architectural design/optimization.

---

> ### Author Response · Authors · 2025-11-20
> **Rebuttal 3/3**
>
> > Is the FlashAttention JVP essential for convergence, or mainly for speed?
>
> FlashAttention JVP is an essential component for the method to train at scale because current official FlashAttention modules do not handle backwards pass through JVP. Naive attention is less than ideal due to speed and memory requirements and it runs out of memory on ImageNet-256x256 with DiT-XL/2 architecture (under PyTorch). The FlashAttention JVP module we propose simultaneously solves the memory, speed, and backwards-through-JVP compatibility problem to enable training at scale. We believe this is a novel empirical contribution to the community in addition to our theoretical contributions.
>
> > Could the method generalize to other architectures or modalities (e.g., U-Nets, text-to-image, or video)?
>
> We have presented our result on CIFAR10 with DDPM++ UNet. To demonstrate TVM’s effectiveness on other modalities, we have successfully scaled it to 10B+ parameter Text-to-Image models. Please see attachment in supplementary materials for samples and prompts. We leave exact training details out as this is beyond the scope of this work. We merely intend to demonstrate the scalability of TVM.
>
> Please let us know if you have any more concerns and we are happy to address them!

---

### Official Review · Reviewer_32jy · 2025-11-01

**Soundness:** 3
**Presentation:** 3
**Contribution:** 3
**Rating:** 8
**Confidence:** 3

**Summary:**

This paper proposes Terminal Velocity Matching (TVM), a single-stage objective that generalizes flow matching by learning ODE integrals between any diffusion timesteps via terminal velocity matching. TVM upper-bounds 2-Wasserstein distance under Lipschitz assumptions. On ImageNet-256×256, TVM achieves 3.30 FID in 1-NFE (SOTA) and a new few-step Pareto frontier (2.49 FID at 2-NFE), with natural n-step interpolation.

**Strengths:**

1. TVM reframes the problem of learning long-horizon ODE jumps as a terminal velocity condition (Eq. 6–7), providing a clean theoretical link between displacement error and velocity matching.
2. Theorem 1 establishes a distribution-level guarantee ($W_2$ upper bound) without requiring multiple particles (unlike IMM).
3. Duality with MeanFlow is clearly articulated: The paper shows MeanFlow matches initial velocity while TVM matches terminal velocity (Appendix E.1), offering a compelling symmetry and explaining why TVM is more stable under random CFG.
4. This paper introduces a custom Flash Attention + JVP kernel to accelerate.

**Weaknesses:**

1. While inference is fast, training cost (FLOPs, GPU-hours) vs. MeanFlow, sCT, or IMM is not reported.

**Questions:**

1. How sensitive is performance to the time sampling distribution $p(t,s)$? The paper uses a biased gap-based scheme — would uniform sampling or curriculum hurt?

---

> ### Author Response · Authors · 2025-11-20
>
> We appreciate the positive feedback from the reviewer. We address the questions below.
>
> > While inference is fast, training cost (FLOPs, GPU-hours) vs. MeanFlow, sCT, or IMM is not reported.
>
> We appreciate the reminder on training cost. It is difficult to accurately estimate the exact FLOPs for JVP forward and backward, we report the GPU-hours here instead. To our best knowledge, IMM is trained with batch size 4096 for 1M iterations. Our reproduction shows runtime of 63s per 100 iterations on 64 GPUs, which amounts to 11200 GPU hours in total. MeanFlow in our reproduction in PyTorch runs 0.76s/iter with 16 GPUs and batch size 256 for 240 epochs on ImageNet, which gives 4433 GPU-hours. Our method backwards through JVP so is a little more expensive per training step. However, we achieve our best results at 300K iterations with batch size 2048 on 64 GPUs. The total GPU hour is 4736. As for sCT, the number is cited from the IMM paper, and the training cost to our best knowledge is not entirely comparable because of early stopping due to instability.
>
> |   | MeanFlow  | IMM  | Ours  |
> |---|---|---|---|
> | GPU-hours  | 4433   |  11200  |  4736 |
>
> Our training cost is marginally higher than MeanFlow due to the additional compute required for backwards pass through JVP, but we converge faster but it is much lower than IMM while achieving superior 1-step results.
>
> > How sensitive is performance to the time sampling distribution? The paper uses a biased gap-based scheme — would uniform sampling or curriculum hurt?
>
> As shown in the ablation studies in Table 3, we experiment with many different time sampling schemes, all of which converge similarly at the same iteration. Some schemes such as gap sampling does indeed achieve superior results which we adopt as default choice. As for uniform sampling scheme, we indeed find it converge slower than logit-normal, as corroborated by vanilla FM training. We present below ImageNet-256x256 FID at 100k iterations.
>
> |   | uniform  | gap  |
> |---|---|---|
> | FID@100k |  5.18   | **3.84**    |
>
> Curriculum training (e.g. Flow Matching warmup) significantly improves convergence speed. We present some preliminary results on ImageNet-256x256 for a TVM model continued training with an SiT trained for 200K steps on FM loss. We achieve low FID already at 25K training iter.
>
> |   | 25K iter  | 50K iter |
> |---|---|---|
> | w/o FM warmup |   44.34  |  4.96   |
> | w/ FM warmup |    **5.44**  |  **4.16**   |
>
> We leave more detailed studies to future work.

---

### Author Response · Authors · 2025-11-20
**Overall Updates**

We thank all reviewers for their insightful comments. We have left specific response to each reviewer. Here we note a few updates to our revision.

> Improved results on ImageNet-256x256

We show that 1-NFE can achieve 3.29 FID and 4-NFE can achieve 1.99 FID surpassing DiT-XL/2 baseline (2.27 FID) while maintaining marginally higher 1-NFE/2-NFE FID.


> New results on ImageNet-512x512

We show state-of-the-art 1-NFE result on ImageNet-512x512 for a model trained from scratch, achieving 4.32 FID, matching sCT-XL (4.33) which has 1.1B parameters. We also achieve 2.94 FID with 4 NFEs, surpassing baseline DiT-XL/2 (3.04 FID),


> Visualizations on the samples

Please check the revisions.

> Runtime and memory analysis

Due to concerns about runtime and memory mainly due to backwards pass through JVP, we have added another section studying the computation cost during training.

> Updated training details

We updated training details in Appendix F in a table to include batch size, training iterations etc.

> Supplementary preliminary Text-to-Image results

Some reviewers also have concern about general applicability of this method beyond ImageNet. Besides updates on 512 resolution, we have conducted additional training on Text-to-Image at 10B+ parameter scale, using pure TVM loss as proposed. Please check supplementary materials for some early samples with 4 NFE. We leave out training details because this is out of scope of this current work. We merely demonstrate the scalability of our approach. Please stay tuned for further announcements regarding TVM.

---

### Author Response · Authors · 2025-12-04
**Rebuttal Summary and Final Remark**

We thank all reviewers for their thoughtful feedback and suggestions for improving our exposition. Below we summarize the main points raised and our responses.

**Reviewer 32jy** *(rating 8)* finds our method technically novel and appreciates the theoretical guarantees under the terminal velocity perspective. The reviewer raises minor concerns about training time and sensitivity to hyperparameters.

We address the training-time concern by reporting total training cost in GPU-hours. Although the per-step cost of our method is higher on average, it converges in fewer iterations, so the overall training time is only about 5% higher than MeanFlow—a marginal increase. We also observe that our method is relatively insensitive to hyperparameter choices: different sampling schedules converge similarly well.

-------

**Reviewer 4c17** *(rating 4)* notes that our method is theoretically elegant, achieves an excellent efficiency–quality tradeoff, and offers practical contributions for large-scale training. The reviewer raises concerns about: (1) intuition for terminal velocity, (2) demonstrations beyond ImageNet-256 (e.g., CIFAR10, ImageNet-512 and text-to-image), (3) whether JVP is a substantive contribution, and (4) the tightness of our theoretical bound.

In the rebuttal, we provide an intuitive explanation of why matching terminal velocity is a sound objective and how it can outperform existing objectives. In our revision, we show that TVM achieves state-of-the-art 1-NFE FID on ImageNet-512 and outperforms diffusion with 4 NFE. In the supplemental material, we further demonstrate that TVM scales smoothly to text-to-image models at the 10B+ parameter scale. For a brief experiment, we demonstrate it works on CIFAR10 and achieve 3.50 1-NFE FID (very suboptimal with time constraints).

We also clarify that our JVP formulation is both novel and central to the paper: backpropagation through JVP has been largely unexplored, and because our algorithm fundamentally requires backward through JVP, our practical kernel implementation is a key part of the contribution. Finally, we note that establishing tightness of the theoretical bound is beyond the scope of this work. Our goal is to use the theory as a principled guide for practical design decisions (e.g., addressing Lipschitz issues in DiT), and we leave a deeper analysis of bound tightness to future work.

-------

**Reviewer ep7Z** *(rating 6)* gives a positive review, highlighting the technical novelty, simplicity, and stability of our method, and finds the release of our JVP kernel particularly useful. The reviewer raises concerns about: (1) runtime costs of backpropagating through JVP, (2) more detailed stability comparisons with MeanFlow, (3) comparison with MeanFlow under improved architectures, and (4) further discussion of design choices such as using EMA as the target.

We address the runtime concern by analyzing both per-step runtime and per-GPU memory usage. We find that naïve MeanFlow (without our kernel) runs out of memory (OOM) and fails to train. With our kernel, TVM is more memory-intensive than MeanFlow with the same kernel, but detaching the JVP gradient yields significantly better runtime. Even without detaching, TVM converges quickly, and the total GPU-hours are only about 5% higher than MeanFlow.

We further show that the performance gains primarily stem from our algorithm rather than architectural tweaks: MeanFlow equipped with our improved DiT architecture still underperforms TVM at 1 NFE. Following the reviewer’s suggestions, we add detailed discussion in the revision explaining that EMA improves stability via variance reduction, that TVM remains stable under randomized CFG settings whereas MeanFlow is more prone to collapse, and that TVM can also be used as a distillation technique.

-------

**Reviewer EXtX** *(rating 6)* finds the method very promising, but raises concerns about whether architectural changes alone account for the performance gains, whether TVM works on CIFAR-10 without guidance, and some missing training details.

In the revision, we provide complete training details, including architecture, optimization hyperparameters, and number of iterations. We show that TVM with and without DiT improvements performs similarly, and that the bulk of the performance improvement is attributable to the TVM algorithm itself. Under limited time, we also experimented on CIFAR-10 and obtained a 1-NFE FID of 3.50, which is competitive with many leading one-step methods, though not fully optimized. As with Reviewer ep7Z, we report per-step runtime and overall training cost to contextualize the efficiency of TVM.

------

Overall, the reviews are positive and supportive of our method. Our revision incorporates all substantive suggestions including: new SOTA results on both ImageNet-256 and ImageNet-512, more comparisons with MeanFlow, and more analyses of key design choices. We hope this work constitutes a meaningful contribution to the community.

---

### Meta-Review · Area_Chair_o7ce · 2026-01-04

**Summary:**

This paper proposes Terminal Velocity Matching (TVM), a principled objective for one- and few-step generative modeling that enforces velocity matching at terminal time. The approach is theoretically grounded via a 2-Wasserstein upper bound and demonstrates strong empirical performance, achieving state-of-the-art efficiency–quality trade-offs on ImageNet-256 and ImageNet-512, with additional evidence of scalability to text-to-image models.

**Reviewer Concerns:**

Reviewers agree on the soundness and technical novelty of the work, though one reviewer raises concerns regarding empirical scope and conceptual clarity. In particular, while the authors argue that matching terminal velocity is fundamental to their approach and empirically beneficial, the paper does not fully articulate why terminal-time velocity matching should improve optimization or convergence relative to initial- or midpoint-based objectives, beyond empirical observations and theoretical framing. This leaves some ambiguity regarding the causal mechanism underlying the performance gains.

Relatedly, several reviewers question the necessity of backpropagation through Jacobian–Vector Products (JVPs). The authors clarify that full JVP backpropagation is essential to enforcing the terminal velocity constraint, but the absence of an ablation detaching JVP gradients leaves the practical necessity of this design choice empirically unverified. These concerns pertain to explanation and empirical completeness rather than correctness.

Despite these open questions, no reviewer identifies fundamental flaws in the formulation, proofs, or experiments. The authors address most substantive concerns through additional results, clearer comparisons, and runtime analysis, strengthening the overall case. While the paper would benefit from a more explicit explanation of the optimization advantages of terminal velocity matching, it presents a meaningful and technically solid contribution to few-step generative modeling.

**Reviewer Scores:**

Reviewer 4c17 may increase the score given that most of the concerns have been addressed. Other reviewer may keep their scores.

---

### Decision · Program_Chairs · 2026-01-26

Accept (Poster)